

# Using metal oxide gas sensors for the estimate of methane controlled releases: reconstruction of the methane mole fraction time-series and quantification of the release rates and locations

Rodrigo Rivera Martinez[1], Pramod Kumar[1], Olivier Laurent[1], Gregoire Broquet[1], Christopher Caldow[3,4], Ford Cropley[1], Diego Santaren[1], Adil Shah[1], Cécile Mallet[2], Michel Ramonet[1], Leonard Rivier[1], Catherine Juery[5], Olivier Duclaux[5], Caroline Bouchet[6], Elisa Allegrini[6], Hervé Utard[6], and Philippe Ciais[1]

[1]Laboratoire des Sciences du Climat et de l'Environnement, LSCE/IPSL, CEA-CNRS-UVSQ, Université Paris-Saclay, 91191 Gif-sur-Yvette, France
[2]Université de Versailles Saint-Quentin, UMR8190 – CNRS/INSU, LATMOS-IPSL, Laboratoire Atmosphères Milieux, Observations Spatiales, Quartier des Garennes, 11 Boulevard d'Alembert, 78280 Guyancourt, France
[3]Clean Energy Regulator, Discovery House, 47 Bowes St, Phillip ACT 2606, Australia
[4]Environment, CSIRO, Aspendale, VIC, 3195, Australia
[5]TotalEnergies - OneTech, Laboratoire Qualite´ de l'Air – 69360 SOLAIZE FRANCE
[6]SUEZ - Smart & Environmental Solutions; Tour CB21, 16 place de l'Iris, 92040 La Défense France

**Correspondence:** Rodrigo Rivera (rodrigo.rivera@lsce.ipsl.fr)

**Abstract.** Fugitive methane ($CH_4$) emission occur in the whole chain of oil and gas production, from the extraction, transportation, storage and distribution. The detection and quantification of such emissions are conducted usually from surveys as close as possible to the source location. However, these surveys are labor intensive, costly and they do not provide continuous monitoring of the emissions. The deployment of permanent networks of sensors in the vicinity of industrial facilities would overcome the limitations of surveys by providing accurate estimates thanks to continuous sampling of the plumes. High precision instruments are too costly to deploy in such networks. Low-cost sensors like Metal oxide semiconductors (MOS) are presented as a cheap alternative for such deployments due to its compact dimensions and to its sensitivity to $CH_4$. In this study we test the ability of two types of MOS sensors from the manufacturer Figaro® (TGS 2611-C00 and TGS 2611-E00) deployed in six chambers to reconstruct an actual signal from a source in open air corresponding to a series of controlled $CH_4$ releases and we assess the accuracy of the emission estimates computed from reconstructed $CH_4$ mole fractions from voltages measurements of these sensors. A baseline correction of the voltage linked to background variations is presented based on an iterative comparison of neighboring observations. Two reconstruction models were compared, multilayer perceptron (MLP) and 2nd degree polynomial, providing similar performances meeting our target requirement on all the chambers when the input variable is the TGS 2611-C00 sensor. The emission estimates were then computed using an inversion approach based on the adjoint of a Gaussian dispersion model obtaining promising results with an emission rate error of 25% and a location error of 9.5 m.



# 1 Introduction

Fossil fuel anthropogenic methane ($CH_4$) emissions related to the production, exploitation and transport of coal, oil and natural gas, account for 35% of global anthropogenic emissions (Saunois et al., 2020). Emissions from natural gas production occur along the chain from extraction, transportation, storage, distribution and use. Emissions estimates reported by inventories rely on information from activity data and emission factors. Emission factors are different between sites, technologies, operating modes and are not stationary, which makes the upscaling of fugitive $CH_4$ emissions highly uncertain (Alvarez et al., 2018). For instance, emissions from the oil and gas supply chain in the US constrained from ground based and aircraft measurements were found to be 60% higher than the EPA inventory (Alvarez et al., 2018). More generally, the characterization of $CH_4$ emissions from complex processes based on static emission factors can be challenged when the best practices are not followed by operators (Riddick et al., 2020).

Atmospheric measurements are increasingly used to detect and quantify $CH_4$ leaks from industrial facilities. The measurements are often interpreted with local-scale dispersion models using atmospheric inversion methods to infer the $CH_4$ source location and emission rates, see e.g. (Kumar et al., 2022). Current approaches generally consist in conducting atmospheric surveys of the enriched concentration plume created by the emitting source. Difficulties are the accessibility to sample the plumes from emitting locations, labor and instrument costs given that surveys currently employ expensive high precision research-level $CH_4$ instruments, such as Cavity ring-down spectrometers (CRDS). Further, surveys do not provide continuous monitoring of the sources (Travis et al., 2020). The deployment and functioning of mini-networks of continuous monitoring sensors for $CH_4$ mole fractions is an alternative to surveys, but the costs of each instrument remain a limitation. Advances in the development of low-cost sensors facilitates the deployment of dense sensors' networks to increase the coverage of a site (Kumar et al., 2015; Mead et al., 2013). Permanent deployment of a network of sensors can overcome limitations in the quantification of leaks and help to better characterize the plumes by limiting the impact of atmospheric transport modelling uncertainties. In addition, the theoretical study of Chamberland and Veeravalli (2006) proved that performance is improved in differentiation of known signals from noise by increasing the sensor density in an area.

In later years, an increase in the interest in low cost and low power sensors to be used in dense networks led to the study of different kinds of sensors to measure pollutants and trace gases like $CO_2$ or $CH_4$. One of the most common low-cost sensors technologies for the detection and quantification of $CH_4$ emissions is metal oxide semiconductors (MOS). MOS sensors are composed of a metal oxide sensing material and a heater ensuring that the sensing material reaches temperatures between 300 to 500 °C. A chemical reaction affects the electrical conductivity of the sensing material in the presence of an electron donor gas such as $CH_4$ (Özgür Örnek and Karlik, 2012). The advantages of MOS sensors are that they are compact and very well suited to long time deployment due to their resilience to extreme weather conditions. However, their sensitivity is affected by environmental parameters (temperature and relative humidity) (Popoola et al., 2018) and VOCs; they also present low accuracy and drift with time (in the form of a decrease in the conductance of the sensing material), requiring periodic re-calibrations, and the need of constant power supply due to the heater material.





The Taguchi Gas Sensors (TGS) commercial MOS from the Figaro® manufacturer, were widely tested in different environments under controlled conditions and field deployment due to their sensitivity to $CH_4$ (Eugster et al., 2020; Eugster and Kling, 2012; Riddick et al., 2020; Collier-Oxandale et al., 2018; Bastviken et al., 2020; van den Bossche et al., 2017). The standard technique to derive a calibration methodology is to collocate these MOS sensors with a high precision instrument used as a reference, then apply empirical equations or data-driven approaches (Eugster et al., 2020; Eugster and Kling, 2012;

Casey et al., 2019; Bastviken et al., 2020; Collier-Oxandale et al., 2018, 2019). In a previous work (Rivera Martinez et al., 2021) we have studied the possibility of using Artificial neural Networks (ANN) to reconstruct the variations of $CH_4$ mole fractions in room air under controlled conditions from three types of Figaro sensors (TGS 2600, TGS 2611-C00 and TGS 2611-E00). A following study (Rivera Martinez et al., 2022) analyzed the potential to reconstruct spikes of $CH_4$ generated on top of ambient air observations, that corresponded to typical signals from leaks at industrial sites, employing two types

of Figaro sensors (TGS 2611-C00 and TGS 2611-E00). That study made a thorough comparison of the performance of five models for the reconstruction of $CH_4$.

The next logical step is to test the performances of the same sensors to reconstruct $CH_4$ from real leaks, and to use the reconstructed mole fractions to quantify emission rates. To our knowledge, only one study attempted to do so. Riddick et al. (2020) quantified emissions of a gas terminal using a Figaro TGS 2600 included in a logger system controlled by an Arduino

Uno. The logging system was located 1.5 m from a point source. To reconstruct $CH_4$ mole fractions from voltage observations, Riddick et al. (2020) developed an empirical equation considering the measured voltage, temperature and the humidity. Then, a Gaussian plume model was used to quantify the emission rate using information from the reconstructed $CH_4$ mole fractions and wind information from a nearby meteorological station. Their estimates of the emissions rates had an average value of 9.6 g $CH_4$ s$^{-1}$ and reached a maximum of 238 g $CH_4$ s$^{-1}$, given corresponding to enhancements of the $CH_4$ mole fractions

between 2 ppm to 5.4 ppm within the plume. Their estimates based on a Figaro sensor were not confronted with high precision instruments nor with an independent knowledge of emission rate.

In this study, we test the ability of a network of several Figaro sensors to reconstruct the $CH_4$ atmospheric enhancements from a series of controlled releases of known magnitudes and duration in open air at a facility called TADI (see Methods), and to infer the emission rate of each release by an inverse modeling approach. The accuracy of the $CH_4$ reconstruction is evaluated

against collocated accurate $CH_4$ measurements from high precision CRDS instruments. The accuracy of the inverted emission locations and rates is evaluated against the known (controlled) location and magnitude using the inversion model of Kumar et al. (2022)

This study builds upon the research conducted by Rivera Martinez et al. (2022) and Kumar et al. (2022), demonstrating the potential for continuous monitoring of $CH_4$ emissions using cost-effective in situ sensors. Drawing from the insights derived

from these two studies, it seeks to address the new challenges associated with the combination of both types of analysis. Firstly, the challenge arises in the deployment and management of Figaro® sensors onsite, an issue not present in Rivera Martinez et al. (2022), as well as extracting $CH_4$ concentrations from measurements that are impacted by more complex perturbations. For instance, the background air in Rivera Martinez et al. (2021, 2022) was less polluted than the one from an industrial site such as TADI. Moreover, the environmental conditions, especially in terms of temperature and moisture, in these previous studies were



smooth and not representative of the real field conditions as encountered in this new study. Secondly, the prescriptive precision
and accuracy targets for $CH_4$ reconstructions outlined in Rivera Martinez et al. (2022) were established as generic targets,
fitting for a variety of data processing strategies intended to quantify emissions from industrial sites. The specific observation
and modelling strategy implemented in Kumar et al. (2022) to localise and quantify point source emissions carries its own set
of precision/accuracy requirements. In particular, this strategy strongly relies on the characterization of gradients across the

measurement stations of concentration averages over time or wind sectors, which makes the derivation of nominal requirements
on the reconstruction of $CH_4$ spikes or time series quite complex. Furthermore, such requirements should be weighed against
the modelling uncertainties associated with the corresponding Gaussian plume model inversions. Ideally, the uncertainties
related to the $CH_4$ data would not significantly add to the total uncertainty when combining them with uncertainties from the
modelling framework. This, however, does not necessarily mean that they should be much smaller than the latter. The direct

comparison of the results obtained in this study with $CH_4$ data derived from the Figaro sensors and those from Kumar et al.
(2022) provides insights into whether this objective is achieved.

Therefore, for 33 controlled releases at the TADI facility, we employed fixed-point measurements from both high precision
CRDS instruments and low-cost TGS. A considerable fraction of the TGS measurements were used for training models to re-
construct $CH_4$ mixing ratios from measured TGS resistance and other variables. When reconstructing the $CH_4$ concentrations,

we proposed a minimum accuracy target for $CH_4$ reconstruction models at 15% of the amplitude of the largest observed excess
within a release. This corresponds to accuracies going from 0.3 ppm for a release causing a maximum excess of 2.4 ppm,
to 18 ppm for a maximum excess of 120 ppm. This accuracy is consistent with the accuracy requirement set in our previous
study where we used TGS sensors to reconstruct $CH_4$ spikes created in a laboratory experiment (Rivera Martinez et al., 2022).
However, the relevance of this target is implicitly re-assessed through the use of the reconstructed time series in the inversion

scheme from Kumar et al. (2022).

The plan of the study is as follows. Section 2 presents the TADI 2019 controlled releases campaign, the logger systems, the
models employed to reconstruct $CH_4$ from TGS data, and the atmospheric inversion approach. The data treatment, comparison
of the models for the reconstruction of $CH_4$ and the inversion results for rates and locations of different releases are analyzed
in section 3. Results are discussed in Section 4, and conclusion are given in section 5.

## 2   Methods

### 2.1   Sampling strategy at the TADI-2019 campaign

In October 2019, TotalEnergies® conducted an experiment with multiple controlled releases at the TotalEnergies Anomaly
Detection Initiative (TADI), to investigate the capability of detection and quantification of different technologies for local
emissions produced on industrial facilities. The TADI test site is designed and operated by TotalEnergies® to test different

technologies and methodologies of detection and quantification of gas leaks in an industrial environment, such as oil and gas
production facilities. The platform is located northwest of Pau, France, with an approximate area of 200 m × 200 m. The site
is equipped with a series of pipes, valves, tanks, , and other equipment commonly found on oil and gas facilities to simulate





'realistic' leaks. The terrain is flat but includes different obstacles that can affect the dispersion of the gases released to the atmosphere. This experiment consisted of 41 controlled releases of $CH_4$ and $CO_2$ covering a wide range of emissions between
0.15 and 150 g $CH_4$ $s^{-1}$ and durations ranging between 25 to 75 minutes. We participated to this experiment to develop and test inverse modelling frameworks within the TRAcking Carbon Emissions (TRACE, https://trace.lsce.ipsl.fr/) program for the estimation of emission location and rates based on $CH_4$ mole fractions from high precision instruments (Kumar et al., 2022). We presented the inversion results for 26 releases from single point sources based on two inversion approaches, one relying on fixed-point measurements, and the other one on mobile near-surface measurements (the latter had already been
documented in (Kumar et al., 2021)). In both cases, the estimates of the emissions relied on $CH_4$ mole fractions from high precision instruments, and on a Gaussian plume model to simulate the local atmospheric dispersion of $CH_4$. The results from Kumar et al. (2022) proved to be relatively good, with an error in the release rate estimates from fixed-point measurements between ~23 to ~30 % and an error in the location of the point sources (within a 40 m × 50 m area) of between 8 and 10 m.

The controlled releases were emitted at different heights up to 6 m above the ground, and inside the 40 m × 50 m ATEX
zone of the TADI facility (see Fig. 1). More information on the site infrastructure and on these experiments in October 2019 are presented respectively in Kumar et al. (2021) and Kumar et al. (2022).

The multiple controlled releases experiment was conducted from October 2, 2019, to October 10, 2019. Our atmospheric sampling configuration for measuring $CH_4$ is shown in figure 2. It consisted of placing 16 sampling lines on the ground connected on one end to air intakes in tripods at heights between 2.75 to 3.50 m around the ATEX zone and on the other
end to a pump flushing at 6 LPM (KNF N811 with PTFE diaphragm). The lengths of the sampling lines varied from 10 m to 100 m connecting each tripod air intake to $CH_4$ measurement instruments located inside a tent. The pump was connected upstream from the high precision instruments (Picarro CRDS or LGR), a chamber containing a series of TGS $CH_4$ sensors, and other sensors measuring environmental parameters such as relative humidity, pressure and temperature. To maintain the inline pressure at atmospheric pressure, a vent was also connected to each sampling line (Fig. 2).

Table 1 summarizes the species measured, and the identifiers of the reference high precision instruments. All reference instruments measured $H_2O$ to provide dry mole fractions of the species. The analysers' sampling frequency ranges between 0.3 to 1 Hz. In a previous study by Yver-Kwok et al. (2015), it was proven that those analysers ensure high precision measurements and a low drift over time, less than one ppb per month. Yet, two calibrations were conducted before and after the campaign. On average 6-7 sampling lines were active for each release, each active line being connected to a high precision instrument and a
TGS chamber. The lines were activated depending on wind direction. The strategy behind the distribution of the tripods around the emitting area and for the inversion was to acquire continuously several measurements points within the plume generated by each release, in addition to one or few measurements points outside the plume (to characterize the background level upon which plumes enhancements can be assessed) for each release regardless of the wind conditions (Kumar et al., 2022).

## 2.2 Low-cost $CH_4$ sensors logger system

Seven chambers were assembled for the campaign. Table 2 displays the sensors included in each chamber, the type of chamber, and the reference instrument with which each chamber was collocated, among other information. Each chamber contained at





**Table 1.** Summary of the species measured by each reference instrument.

| Serial number / Code | Identifier | Species measured |
| --- | --- | --- |
| CFKADS2286 / Picarro 1 | Picarro CRDS G2401 | $CH_4$, $CO_2$, CO |
| CFKADS2301 / Picarro 2 | Picarro CRDS G2401 | $CH_4$, $CO_2$, CO |
| CFKADS2194 / Picarro 3 | Picarro CRDS G2401 | $CH_4$, $CO_2$, CO |
| CFKADS2131 / Picarro 4 | Picarro CRDS G2401 | $CH_4$, $CO_2$, CO |
| CFIDS2067 / Picarro 5 | Picarro CRDS G2201-i Isotopic | $^{13}CH_4$, $^{12}CH_4$, $^{13}CO_2$, $^{12}CO_2$ |
| CFIDS2072 / Picarro 6 | Picarro CRDS G2201-i Isotopic | $^{13}CH_4$, $^{12}CH_4$, $^{13}CO_2$, $^{12}CO_2$ |
| LGR MGGA | Los Gatos Micro-portable Greenhouse gas analyzer | $CH_4$, $CO_2$ |

least three TGS with voltage measurements sensitive to $CH_4$ and two other sensors measuring relative humidity/temperature and pressure/temperature. All these sensors were inserted inside an acrylic/glass or steel/glass chamber with volumes of 100 ml and 120 ml, respectively. The sensitivity of TGS was controlled by a load resistor connected in series to the sensor (Figaro®, 2013, 2005), two values of resistor were used, 5KΩ and 50KΩ (see table 2 for details). An AB Electronics PiPlus ADC board mounted on a Raspberry Pi 3B+ recorded the voltage on the load resistor providing observations every 2s. This voltage is used for the characterization and reconstruction of the $CH_4$ signal. Consistency was observed between the two TGS 2611-E00 sensors installed on chamber E, and only one sensor of this type is used in this study.

Measurements of environmental parameters from the other chambers, besides chamber E, had data gaps for extended periods or bad recordings occurring at releases and were not included. This study focuses on reconstructing $CH_4$ using data from TGS 2611-C00 and TGS 2611-E00 from chambers A, C, D, E, F, and H. Data from TGS 2600 were discarded since this sensor did not respond to most of the $CH_4$ peaks during the releases (see Figure A1).

## 2.3 TADI controlled releases and meteorological data

A total of 41 controlled releases were conducted during the seven days of experiment, with release durations varying between 25 to 75 minutes. Because low wind conditions ($Ur < 0.6$ m s$^{-1}$) are not suitable for the atmospheric inverse modelling, six releases corresponding to such low wind conditions have been excluded for the inversion modelling here such as in Kumar et al. (2022). However, they are used in the training of the $CH_4$ reconstruction models. There was no TGS measurements during the five releases corresponding to the last day of the campaign. Two largest releases produced high $CH_4$ mole fraction plumes that affected the amplitude measured by the TGS sensors on which it was not possible to distinguish large spikes from medium and small ones on the measured voltage (see fig A3) and they are removed from the study. One release was aborted due to technical problems at the site and is as well removed from this work. This study is thus focused on 33 out of the 41 controlled releases. A summary of these releases is shown in Table A1, where an 'x' indicates invalid data measured by the chambers. This invalidity is due to some releases producing small peaks over the background signal (with enhancements of less than 4 ppm), which the TGS sensors were unable to detect.



**Table 2.** Summary of the specifications of the chambers, the tripods to which each chamber was connected, the captured releases and the reference instrument collocated with each chamber.

| Chamber | Figaro TGS sensors | Load resistor ($\Omega$) | Other sensors | Chamber type | Tripod | # of measured releases | Reference instrument |
|---------|-------------------|--------------------------|---------------|--------------|--------|------------------------|---------------------|
| A | 2600<br>2611-C00<br>2611-E00 | 50K | DHT22<br>BMP280 | Acrylic/glass | 1, 4, 6<br>8, 9, 10<br>11, 14, 15 | 28 | Picarro CFKADS2286 |
| C | 2600<br>2611-C00<br>2611-E00 | 50K | SHT75<br>BMP280 | Acrylic/glass | 2, 7, 9<br>14, 15, 16 | 24 | Picarro CFIDS2072 |
| D | 2600<br>2611-C00<br>2611-E00 | 5K | SHT75<br>BMP280 | Steel/glass | 2, 3, 9<br>10, 11, 12<br>13, 16 | 14 | Picarro CFKADS2301 |
| E | 2600*<br>2611-C00*<br>2611-E00* | 5K | DHT22<br>SHT75<br>BMP180 | Steel/glass | 1, 3, 4<br>5, 10, 11<br>12, 13, 16 | 24 | Picarro CFKADS2131 |
| F | 2600<br>2611-C00<br>2611-E00 | 50K | SHT75<br>BMP280 | Acrylic/glass | 2, 3, 4<br>10, 11, 12<br>13, 14, 15 | 25 | Picarro CFKADS2194 |
| H | 2600<br>2611-C00<br>2611-E00 | 50K | SHT75<br>BMP180 | Acrylic/glass | 4, 5, 6<br>7, 12, 13<br>14, 15 | 22 | LGR MGGA |

* Two versions of each type.

The protocol followed in the selection of the releases used in the training and test set for the reconstruction models is explained in section 2.6. A meteorological station was installed on the TADI platform by TotalEnergies® with a sonic 3D anemometer at 5 m height above the ground surface (see Fig. 1), providing 1-minute averages of wind speed (Ur), wind direction ($\theta$) and of the standard deviation of wind speed on the three axes ($\sigma_u$, $\sigma_v$ and $\sigma_w$) amongst other parameters. The data of turbulence and meteorological conditions are used in the dispersion model. Table 3 gather general information for each of the 33 controlled release during which we have valid TGS measurements: the duration of the release, the actual release rate, the average wind speed over the duration of the release and an indication showing if it was selected for the inverse modelling.





**Table 3.** Summary of the information for the controlled releases with single $CH_4$ point sources during the TADI 2019 campaign. Rows in gray shows the releases with low wind speed conditions.

| Release number | Duration (hh:mm) | Emission rate ($Q_s$ ($g\ s^{-1}$)) | Average wind speed ($U_r$ ($m\ s^{-1}$)) | Used in the atmospheric inverse modelling |
|---|---|---|---|---|
| 1 | 00:58 | $CH_4$: 10 | 2.76 | No |
| 2 | 00:32 | $CH_4$: 1 | 3.31 | Yes |
| 3 | 00:33 | $CH_4$: 0.5 | 3.56 | No |
| 4 | 00:33 | $CH_4$: 5 | 3.91 | No |
| 5 | 00:35 | $CH_4$: 3, $CO_2$: 85 | 0.65 | Yes |
| 6 | 00:39 | $CH_4$: 0.5 | 0.45 | No |
| 7 | 00:46 | $CH_4$: 5.0 | 0.80 | No |
| 8 | 00:50 | $CH_4$: 0.5 & 0.75 & 0.5 * | 1.41 | No |
| 9 | 00:38 | $CH_4$: 1, $C_2H_6$: 0.5 | 1.46 | Yes |
| 10 | 00:38 | $CH_4$: 0.5 | 2.17 | Yes |
| 11 | 00:30 | $CH_4$: 0.16 | 2.39 | No |
| 12 | 00:46 | $CH_4$: 1 | 0.93 | Yes |
| 13 | 00:44 | $CH_4$: 0.2 | 0.26 | No |
| 14 | 00:55 | $CH_4$: 0.5 & 1.0 * | 0.07 | No |
| 15 | 01:01 | $CH_4$: 2 | 3.50 | No |
| 16 | 00:44 | $CH_4$: 2 | 1.83 | No |
| 17 | 00:50 | $CH_4$: 4 | 1.45 | No |
| 18 | 00:48 | $CH_4$: 0.3 | 0.13 | No |
| 19 | 00:40 | $CH_4$: 2.0 | 0.41 | No |
| 20 | 00:58 | $CH_4$: 2 & 4 * | 0.47 | No |
| 21 | 00:44 | $CH_4$: 1 | 1.31 | Yes |
| 22 | 00:33 | $CH_4$: 1, $C_2H_6$: 0.2 | 1.11 | No |
| 23 | 00:50 | $CH_4$: 2 | 1.84 | No |
| 24 | 00:43 | $CH_4$: 150 | 2.63 | No |
| 25 | 00:35 | $CH_4$: 5 | 3.12 | Yes |
| 26 | 00:48 | $CH_4$: 0.4 | 2.73 | Yes |
| 27 | 00:37 | $CH_4$: 0.5 | 3.12 | No |
| 28 | 00:45 | $CH_4$: 0.5 & 0.5 * | 1.04 | No |
| 29 | 00:44 | $CH_4$: 0.6 | 1.07 | Yes |
| 30 | 00:44 | $CH_4$: 1 | 1.51 | No |
| 31 | 00:24 | $CH_4$: 2 | 1.70 | No |
| 32 | 00:34 | $CH_4$: 4 | 3.58 | Yes |
| 33 | 00:45 | $CH_4$: 2 | 2.49 | Yes |

* Multiple source releases.



## 2.4 Reconstruction of spikes in $CH_4$ mixing ratios caused by the releases

The chambers full of TGS sensors captured different portions of the plume with variations at high frequencies due to the distribution of the tripods with regards to the variable wind direction and due to the turbulence. The typical signal measured

by the chambers is a series of spikes, ranging between 1 and 15 minutes, corresponding to the plume lying over a slowly varying background signal associated to remote emissions. The targeted signal is that of the difference between the spikes and the background (Kumar et al., 2022). As an example, Figure 3 shows 1-minute averages $CH_4$ mole fractions measured by the reference instruments and the voltage from the TGS 2611-C00 at six tripods during release 25 ($Q_s = 5$ g s$^{-1}$). We can observe that $CH_4$ of the reference instrument and TGS voltage show good consistency at this temporal resolution. Chamber A, C and

D were in the trajectory of the plume or very close to it measuring peaks up to 30 ppm, chambers E and F only captured one peak of $\sim$10 ppm and chamber H one large peak of 30 ppm. The mean wind speed during this release was of 3.12 m s$^{-1}$ and the wind direction had little variations, ranging between 270° to 272°.

TGS sensors are known to be sensitive to variations of $H_2O$ and T, affecting mainly the reconstruction of $CH_4$ baseline, and thus the characterization of peaks above this baseline (Rivera Martinez et al., 2021, 2022). Two approaches can be used to

195 correct the effect of variable $H_2O$ and T on the TGS signals baseline and separate the spikes from the baseline data in the time series. The first one is the use of information from $H_2O$ and T to correct the TGS baseline signals correspond to these drivers. The second approach is to detect the voltage peaks associated to $CH_4$ spikes and derive a baseline with a linear interpolation on non-peak voltages. For some chambers due to logging system faults, we lost $H_2O$ and T data and the corresponding gaps in the $H_2O$ and T time series prevent us from defining a correction model. Therefore, in this study we have employed the second

approach. To justify our choice, we have trained a multilinear regression model to determine a baseline signal on TGS 2611-C00 from Chamber E corresponding to $H_2O$ and T. The regression model was trained on using observations from midnight to 6:00 in the morning on the first day and we attempted to reconstruct baseline variations of TGS voltage from observations comprised between 18:00 to midnight on the same day. The results of the multilinear model are presented on Figure A2 (in addition to the derived baseline when using the second approach). The second approach which produce a better detection of the

baseline signal is also shown (see Fig A2) where we do not need a training set or environmental variables because it consists in the detection of peaks based on an iterative process on fixed rolling windows and a comparison with neighbouring observations.

To reconstruct $CH_4$ mole fraction, we calibrated empirical models that derive relationships between TGS voltage and other input variables and true $CH_4$ observed by the high-precision instruments. The models are calibrated (training) and evaluated (testing) using two independent subsets of the data. Following the widespread practice in the training of data-driven models to

210 standardize the input variables to prevent difference in the range of magnitudes from conditioning the determination of model parameters, we applied a robust transformation consisting in removing the median and dividing the observations by their 1-99th quantile range. We selected the two reconstruction models that gave the best performances in our previous study (Rivera Martinez et al., 2022), namely a polynomial regression and a Multilayer Perceptron (MLP) model, described below.

Second-degree polynomials have proven to be robust to derive relationships between the TGS voltage signal related to spikes

and the corresponding $CH_4$ concentration (Rivera Martinez et al., 2022). Its formulation is of the form:





$$\hat{y}_{CH_4}(x_1) = \beta_0 + \beta_1 x_1 + \beta_2 x_1^2 \tag{1}$$

Where $\hat{y}_{CH_4}$ is the predicted $CH_4$ concentration, $x_1$ is the Corrected voltage of the TGS after removing the effects of the baseline.

Artificial neural networks have been widely used to derive non-linear relationships between predictors and independent
variables in many applications, as a universal approximator method (Hornik et al., 1989) and for their generalization capabilities (Haykin, 1998). In previous studies (Casey et al., 2019; Rivera Martinez et al., 2021, 2022) ANN was employed to derive $CH_4$ concentrations from TGS observations on different sampling configurations (field and laboratory conditions) with good agreement between the reference observations and the outputs produced from the models.

The simplest architecture of an ANN is the multi-layer perceptron (MLP), conformed of a series of units (neurons) in
fully connected layers. The inputs of any unit will be the weighted sum of the outputs of the previous layer, to which an activation function (ReLU, tanh, etc.) is applied. As a machine learning approach, it requires a training basis to learn the relationships, adjusting the weights of its connections, between the inputs and outputs using an iterative process known as optimization. Problems of MLP models are either underfit of data, producing a high error on the train set which can be mitigated with a sufficiently large network, or overfitting, producing a high test error when they cannot generalize to new
examples. Regularizations terms and early stopping techniques are helpful to prevent overfitting (Bishop, 1995; Goodfellow et al., 2016).

Here, we have trained the MLP model using the Adam optimizer (Kingma and Ba, 2014; Géron, 2019) resulting in 50, 10 and 5 units per layer with ReLU as the activation function for the hidden units. A regularization factor of $\alpha$=0.05 and early stopping was used to prevent overfitting.Three configurations of the input variables were tested: i) only with the TGS 2611-C00, ii) only
with the TGS 2611-E00, and iii) with both TGS sensors at the same time. The results are shown in section 3.2.

## 2.5 Metrics for evaluation of the reconstruction

To assess the performance of the models to provide dry $CH_4$ concentration enhancements (above the background) from voltage observations of the low-cost sensors we use a normalized root mean square error (NRMSE) per release, including information from the spikes and the background occurring in the duration of the release, defined in equation 2, the RMSE being weighted
by the inverse of the maximum peak present in the release:

$$NRMSE = \frac{\sqrt{\frac{\sum(y_i - \hat{y}_i)^2}{n}}}{h_{max}} \tag{2}$$

where $y_i$ is the actual concentrations (provided by the high precision instrument),$\hat{y}_i$ the predicted concentration, n the number of observations present in the release and $h_{max}$ the amplitude of the maximum peak present in the release after removing the background. The normalization allows to compare the performances across the different releases.





As mentioned earlier in section 2.4, the target signal on this study is of $CH_4$ enhancements above the atmospheric background. We obtain this signal by subtracting the raw signal of the release from an inferred baseline computed using the pic detection algorithm and a linear interpolation. We consider as an acceptable notional target error for the reconstruction models to be under the 15% of the amplitude of the maximum peak inside the release, this error corresponds to a NRMSE $\leq 0.15$ ppm.

## 2.6    Selection of the training and test subsets for the reconstruction of $CH_4$ mole fractions as input of the atmospheric
inversion of emissions

Defining the appropriate training set is important to allow reconstruction models to derive sufficient information to generalize and obtain good performances in the test set. As well, the test set should be chosen to allow evaluating the performances of the models under a wide variety of conditions. Regarding the inverse modelling, in order to provide a meaningful assessment of the estimation of emission rates and locations, inversion should be conducted using reconstructed $CH_4$ mole fractions that are
not from the training data set to avoid introducing bias in the evaluations of errors. Furthermore, depending on magnitude of release rates, the atmospheric turbulence, and the locations/distances of the downwind active tripods from the emission sources, the six chambers did not measure $CH_4$ mole fractions in all the releases, therefore a separate training and test set needs to be defined for each chamber.

The previous considerations constrain the selection of the training and test sets from the data of each chamber. The test set of
the releases for inversions was defined based on two criteria: 1) the releases which have the reconstructed $CH_4$ mole fractions by at least three chambers simultaneously, and 2) the releases corresponding to the more favourable wind speed conditions ($Ur \geq 1.4$ m s$^{-1}$) for inversions. We determined seven releases that meet these considerations: release #2, #9, #10, #25, #26, #32 and #33. Because this test set was not sufficiently large for all the chambers, we decided to increase it by data from four more releases with low wind speed conditions ($0.65 \leq Ur \leq 1.31$) (release #5, #12, #21 and #29). This selection led to a test set of
40% of the releases. All remaining data were used as a training set (Table 4). The reconstruction models are trained and tested only once per chamber following the distribution of the releases from table 4.

### 2.7    Atmospheric inversion of the release locations and emission rates

Our derivation of the release location and rates relies on the inversion framework developed and tested by Kumar et al. (2022) on the measurements of the high precision instruments. This framework uses adjoint of a Gaussian plume model to simulate
the sensitivity of the $CH_4$ mole fraction enhancements above the background at a measurement location to the emissions at all potential source locations. For each release, the optimal horizontal and vertical location and rate are derived based on the minimization of the root sum square (RSS) misfits between averages of the observed and simulated $CH_4$ mole fraction enhancements above the background. The bins of the measurements and of the simulated mole fractions for the averages correspond to sectors of wind directions of equal ranges during the release. The optimal release location and rates are searched
simultaneously, looping on a finite but large ensemble of potential locations, using an analytical formulation of the problem to derive the optimal rate and corresponding RSS misfits for each potential location and then identifying the optimal location and rate providing the smallest RSS misfits. The 40 m × 50 m (horizontally) × 8 m (vertically) volume above the ATEX zone



**Table 4.** Summary of the releases included in the training set and test set of the $CH_4$ reconstruction models. The mixing ratios modelled for the test set are also used as input of the inversion model to infer the emission rate of $CH_4$ and their location.

| Chamber | Releases in the training set | Releases in the test set | Number of releases in the training set | Number of releases in the test set | Percentage of releses in the training/test set |
|---------|------------------------------|--------------------------|----------------------------------------|------------------------------------|-------------------------------------------------|
| A | 6, 7, 8, 11, 14, 15, 16, 17, 18, 19, 20, 24, 27, 28, 30 | 2, 5, 9, 10, 21, 25, 26, 29, 32 | 15 | 9 | 62.5 % / 37.5 % |
| C | 14, 15, 17, 18, 19, 20, 22, 24, 27, 28, 30, 31 | 9, 10, 21, 25, 26, 29, 32, 33 | 12 | 8 | 60 % / 40 % |
| D | 6, 7, 8, 13, 14 | 5, 9, 12, 25 | 5 | 4 | 55.5 % / 44.5 % |
| E | 3, 4, 6, 7, 8, 13, 14, 19, 20, 22, 23 | 2, 5, 9, 12, 21, 25, 26, 32, 33 | 11 | 9 | 55.5 % / 44.5 % |
| F | 3, 4, 6, 7, 8, 13, 14, 15, 18, 19, 20, 22, 24 | 2, 5, 9, 10, 12, 21, 25, 29 | 13 | 8 | 62 % / 38 % |
| H | 1, 3, 4, 13, 14, 18, 19, 20, 23, 24, 28, 30 | 2, 21, 25, 26, 29, 32, 33 | 12 | 7 | 63 % / 37 % |

is discretized with a high resolution (1 m × 1 m horizontally and 0.5 m vertically) 3D grid to define the finite ensemble of potential locations. The inversion exploits the change of wind direction during a release and the corresponding variations and
spatial gradients in average mole fractions respectively at and between the different measurement locations crossed by the plumes to triangulate the release location. The amplitude of the enhancements directly constrains the release rate estimate.

The Gaussian model and its adjoint are driven by averaged wind directions and averaged turbulence parameters derived from 3D sonic measurements, using the same bins for these averages as for the mole fractions. Those bins are defined during each release based on 1-min averaged wind directions. These bins partition the lower and upper range of potential wind directions,
and they have equal width in terms of range of wind directions. The total number of bins during this initial partition is defined as the rounding integer of the division of the release duration (in min) by approximately 7 min. However, only bins gathering at least four 1-min averages are retained. The aim is that the mole fraction and meteorological averages are representative of a timescale that is long enough for use in or comparison to the Gaussian model. Depending on the releases, the number of bins ranges between 2 and 7.





Here, we slightly revise the reference computations of release location and rate estimates based on the high precision instruments from Kumar et al. (2022). Indeed, in order to compare the release location and rate estimates from such a reference and the one derived here based on the TGS sensors, we restrain the set of high precision observations that are used in the reference computation to the station and time corresponding to the data availability from the TGS sensors.

## 3    Results

### 3.1    Pre-processing of the data from the low-cost $CH_4$ sensors

Original observations with a time step of 2 s were resampled to 5 s. We corrected the time offset corresponding to delays of the air travel through the air intake from the tripods to the instruments, time delay from synchronization between analysers and chambers. Also, we removed invalid data produced by the logging system on each chamber. The baseline correction was then applied for each sensor chamber considering the entire campaign. As an illustration of the impact of the baseline correction

Figure 4 shows the signal corresponding to one release for the chamber A after these pre-processing steps. The corrected signal in the TGS voltage measurements showed better agreement with the reference between the occurrences of spikes and phases.

### 3.2    Reconstruction of $CH_4$ mole fractions

Due to the diversity of the releases, environmental conditions, distribution of the tripods and selection of the training and test sets for each chamber, there is no single release that can be viewed as representative for the test set across the chambers. Yet,

we chose release #25 as an example of the signal measured across the chambers and the reconstructed signal for each chamber using the MLP model (Figure 5) and the 2nd order polynomial model (Figure 6), for each chamber we shown the reconstructed $CH_4$ mole fractions estimated using only the type C sensors (red), the type E sensor (yellow) and both sensors used as inputs for the models at the same time (green).

    We found that the MLP and 2nd degree polynomials gave similar performances across the releases regardless of the chamber

used for the $CH_4$ reconstruction. For two releases on chamber A (release #10 and #26, see Fig A6 and A9 for MLP model and A11 and A12 for the polynomial model respectively) where amplitudes are below 10 ppm, the polynomial model provides a noisy signal as output regardless the configuration of the inputs used. There were however some cases on which the polynomial model produced better outputs than the MLP, for example the four releases on chamber D where MLP model produced a systematic underestimation of the reconstructed $CH_4$ on the three configurations of inputs.

Regarding the TGS types, the type C sensor gave better reconstructions than the type E or both types used as the same time as inputs for the model. The reconstruction of $CH_4$ with the type E sensor shows phasing errors in the form of a slow decay after large spikes. In addition, there are some cases where type E sensors showed a response whereas no spikes were measured by the reference instrument. For example, release #9 (Figure A5 and A10, for the MLP and the polynomial model respectively) of chamber D shows few spikes between 10 to 30 ppm reconstructed from data of the type E sensor with the polynomial model

which are not present on the reconstructed data from the type C sensor. Using Type C and E sensors at the same time as training





data for models produced outputs closer to models trained only with type C sensor. Some cases of reconstruction with MLP model produced a saturation of the outputs (release #9, #12 and #25 for chamber D (Figure A5, A7 and 5), release #21 for chamber H (Figure A8)) or a systematic bias (releases #2, #10 and #26, see Figure A4, A6 and A9). For releases with peaks' amplitudes above 40 ppm a systematic underestimation is observed regardless the model or the sensor's type used as input.

On figure 7, we present a summary of the performance of the reconstruction of the signal on the test set, given the NRMSE error defined in eq. 2. All chambers have reached our target error of NRMSE ≤ 0.15 ppm, except for Chamber A with the polynomial model using as input the type E sensor and the MLP model for chambers A and C as well for the type E sensor. With a stricter target requirement of NRMSE ≤ 0.1 ppm, only Chamber H met the target error regardless of the model or sensor used. Performances are similar when using the type C sensor as input regardless the model across all the chambers. When used
both types at the same time as input, the 2nd degree polynomial provide better reconstruction than the MLP specially on chambers C, D and H (NRMSE = 0.09, 0.09 and 0.04 ppm for the polynomial model and 0.11, 0.13 and 0.07 ppm for the MLP). Chamber D, where there is little training data available, produced a systematic lower error with the polynomial model than with the MLP regardless the input variable used.

In summary, the model used in the reconstruction is important only for the cases where little information is available for the
training. This was the case for chamber D where the polynomial model provides better performances than the MLP model. We also found that Type C sensors produced a better reconstruction of $CH_4$ spikes than Type E ones, and a combination of data from both types of sensors did not improve the reconstruction producing similar outputs than the other types.

### 3.3    Release rate and location estimates based on the observations from the TGS sensors

Averages of mole fractions enhancements above the background and their spatial gradients are displayed for release #25 in
figure 8. The figure compares the values of reconstructions from the low-cost sensors (with the MLP model; see figure A14 for the values corresponding to the polynomial model), with the high precision measurements, and of the simulations resulting from the inversions assimilating either the reference high precision data or the reconstructions from the low-cost sensors. Since the best reconstruction performances were obtained when using the type C sensor, the inversion results presented here are based on the reconstructions from those sensors only. For the release #25, used as an example here, the procedure to define average
values per wind sectors has resulted in four bins of wind sectors with an approximate size of $10°$. Average mole fractions are derived from the six chambers. To simplify the numbering when mentioning the reference instrument or the TGS, we refer to the chamber identifier X (REF-X and TGS-X respectively, with X the name of the chamber).

In general, the observed spatial $CH_4$ gradients between the different stations are similar when considering the reference measurements and the estimates of the TGS, except for few cases where the reference is more consistent to the expected signal.
For example for release 25 (see Fig. 8) observed gradient from TGS-D data underestimate the actual gradients given by REF-D for $\theta = 308.3°$ and overestimate them for $\theta = 279.2°$, where $\theta$ is the average direction of the wind sector.

The modelled average mole fractions enhancements and thus the modelled gradients assimilating reference data are very close, in general, to the ones from these reference data, although some discrepancies can occur, e.g., for release #25, for REF-H with $\theta = 279.2°$, REF-C with $\theta = 301.4°$ and $\theta = 289.1°$ and REF-A with $\theta = 301.4°$ and $308.3°$. For most of the cases, the





modelled gradients assimilating the TGS data are closer to the modelled gradients assimilating the reference data than to the observed TGS ones. In addition, the observed TGS data, for some cases, is closer to the observed reference one than to the modelled gradients assimilating either reference or TGS data, highlighting the higher impact of the model error on the inversion than the reconstruction error of $CH_4$ mole fractions.

Figure 9 shows the comparison of the emission rate estimates with corresponding errors, and of the location errors for the 360 different inversions across the eleven releases. In this figure, estimates assimilating $CH_4$ mole fractions from the TGS using the reconstruction with the MLP models (see Figure A14 for the results when assimilating the reconstruction based on the 2nd degree polynomial model).

Regarding the release rate estimates, those from inversions assimilating the reference mole fractions bear an average error of 30% and those from the inversion assimilating data from the TGS sensors bear an average error of 25%.

In the case of the estimation of the release location, the assimilation of the reference data produces a slightly smaller average error location of 7.86 m ($\sigma$ = 5.47 m) compared to 9.49 m ($\sigma$ = 4.58 m) from the assimilation of TGS data. For five releases (#2, #10, #12, #25 and #26) the assimilation of reference data yields a better estimate of the location and for one release (#21) both inversions yield similar location errors.

In general, estimates of the emission rate (see fig 9a) from reference data and TGS data are similar. For three releases 370 (#12, #25 and #32), we observe large errors in the estimate of the release rate. Inversion assimilating TGS data or reference data highly underestimate the rate for release #5 (1.41 and 1.34 g $CH_4$ s$^{-1}$ respectively, with an actual emission rate of 3.0 g $CH_4$ s$^{-1}$) and strongly overestimate the rate for release #32 (5.14 and 6.55 g $CH_4$ s$^{-1}$ respectively, with an actual emission rate of 4.0 g $CH_4$ s$^{-1}$). Reference data provide a slightly better estimation of the location of releases than the TGS. Only for releases #29 and #33, the inversion assimilating TGS observations provide a slightly better location of the source. 375 Conversely, for releases #2, #12, #25 and #26, the location error from the inversion assimilating TGS observations is almost double than the one of the reference. The errors on the emission rate estimate from both inversions was smaller than 30% for most of the releases, except on four cases, where errors reached 80% for the inversion assimilating TGS data and 65% for the inversion assimilating reference data, respectively. There were two cases, the release #26 and #33, when the inversion assimilating TGS observations produced a much lower error (2.5% and 5.3% respectively) in the quantification of the emission 380 rate than the inversion assimilating reference observations (20.9% and 22.7% respectively). The fact that the assimilation of the TGS reconstructed $CH_4$ data can yield better results than when using accurate $CH_4$ mole fractions measured by the reference instrument highlights the impact of the transport model error (associated to the simulation of the average mole fractions with the Gaussian model) in the inversion process. These errors dominate the resulting errors in the estimates of the release rate and location when assimilating the reference data Kumar et al. (2022). They appear to have a weight larger than that of the errors 385 in the reconstructed mole fraction from TGS data when assimilating these data.





**Table 5.** Comparison of the emission rate estimates ($Qe$), location error ($El$) and relative error on the rate estimates for the inversions assimilating the reference data and the reconstruction of the $CH_4$ from the TGS low-cost sensor based on the MLP model

| Release N° | Actual emission (g $CH_4$ s$^{-1}$) | Reference | | | TGS | | |
|---|---|---|---|---|---|---|---|
| | | $Q_e$ (g $CH_4$ s$^{-1}$) | $E_l$ (m) | error (%) | $Q_e$ (g $CH_4$ s$^{-1}$) | $E_l$ (m) | error (%) |
| 2 | 1.0 | 1.10 | 5.26 | 10.8 | 0.89 | 12.40 | 10.1 |
| 5 | 3.0 | 1.34 | 21.57 | 55.2 | 1.41 | 19.55 | 52.8 |
| 9 | 1.0 | 0.88 | 14.29 | 11.9 | 1.11 | 12.78 | 11.8 |
| 10 | 0.5 | 0.40 | 9.29 | 18.9 | 0.42 | 10.80 | 14.8 |
| 12 | 1.0 | 0.34 | 3.08 | 65.7 | 1.84 | 7.15 | 84.9 |
| 21 | 1.0 | 0.63 | 3.61 | 36.1 | 0.66 | 3.61 | 33.8 |
| 25 | 5.0 | 4.61 | 4.57 | 7.8 | 5.41 | 10.02 | 8.2 |
| 26 | 0.4 | 0.31 | 5.10 | 20.9 | 0.39 | 10.10 | 2.5 |
| 29 | 0.6 | 0.45 | 3.40 | 24.5 | 0.43 | 2.34 | 28.3 |
| 32 | 4.0 | 6.55 | 10.55 | 63.8 | 5.14 | 10.28 | 28.6 |
| 33 | 2.0 | 2.45 | 5.77 | 22.7 | 2.10 | 5.37 | 5.3 |
| **Average error** | | | 7.86 | 30.7 | | 9.49 | 25.5 |
| $\sigma_{error}$ | | | 5.47 | 20.3 | | 4.58 | 23.6 |

## 4  Discussion

Our study showed the capability of the signal from metal oxide sensors to produce estimates of the emission rate and location from controlled $CH_4$ releases typical of those expected from leaks in industrial facilities. The used baseline correction algorithm allows to extract the variations of voltages from the TGS signal related to the high frequency variation of the plume across

the different sensors' inlets. We compared the performances of two models, 2nd degree polynomials and MLP, to reconstruct $CH_4$ mole fractions during the controlled releases for three configurations of inputs. The reconstructed $CH_4$ mole fractions were used as input to an inversion modelling framework to estimate the emission rate and location for each release. Results of inversions assimilating TGS data were compared with those assimilating reference (CRDS) data.

The correction of baseline in TGS sensors assumes that the targeted signal measured by the sensors corresponds to a series

of spikes at high frequency produced by the plume reaching and leaving the inlet tube of the sensors, due to the atmospheric turbulence and high frequency variations of the wind. Our approach of deriving a baseline signal from observations surrounding the spikes in an iterative process, offers a suitable alternative to correct the TGS observations when little or insufficient information is available to derive a baseline correction model (e.g. from observations of $H_2O$ and temperature). This approach is interesting for conditions when the environmental parameters are highly variable or models does not dispose of sufficient

observations to derive robust relationships to correct the effects of environmental variables on the sensors' baseline signal.



This corresponds well with the measurements presented in this study. However, in some cases, the plume can touch the inlet tube of the sensors during a prolonged period producing a signal not only having high frequency spikes but also continuous varying enhancements above the background. For those cases, this method would not be able to distinguish the enhancements on sensors' voltage corresponding to the $CH_4$ plume from the background and then we need to reconsider the derivation of a

baseline based on environmental parameters ($H_2O$ and T). Regarding the type of sensor used in the reconstruction of $CH_4$ mole fractions, we obtain best performances using only with Type C sensor as input for the models. The fast decay observed on the reconstructed $CH_4$ after the spikes was attributed to the response time of the TGS sensor. The slow decay observed on Type E sensors was probably due to a combination of the response time and the carbon filter added on top of the sensitive material to improve the selectivity of gases. Concerning the reconstruction models, the polynomial and the MLP models in general

produced equivalent results with few differences. It confirms our previous study (Rivera Martinez et al., 2022) in which we observed that the performances of models to reconstruct $CH_4$ mole fractions were mainly driven by the type of sensor used, rather than from the model chosen for the reconstruction. With a low content of information (only few spikes, limited range and variability of the spike magnitude, frequency and duration) in the training set (e.g. when reconstructing the $CH_4$ mole fractions of chamber D), the 2nd degree polynomial provides more accurate estimates than the MLP. This is probably due to

the distribution of the data in the training set that MLP used to compute its parameters, which does not represent the same range of variations than the one in the test set. For spikes with enhancements under 5 ppm, the MLP model with the Type C sensor signal as input, produced a more accurate reconstruction than the Type E or both sensor's types when used as inputs at the same time. The noise present in the voltage signal on some releases, for example release #26 on chamber A, were not correctly removed in the reconstruction with the Polynomial model. However for the type C sensor, the MLP model reduces

the noise on the signal producing a more accurate reconstruction.

Regarding the inversion of emission rates and locations using the Gaussian plume model framework developed by Kumar et al. (2022), we obtained good estimates and performances with the reconstructed time series of $CH_4$ spikes from voltage measurements of TGS sensors and the results are comparable to those obtained when assimilating the reference data. We observed that the simulated gradients of the Gaussian model assimilating observation from the TGS chambers were close to

the simulated gradients of the reference inversions (assimilating high precision measurements), even if the observed gradients were sometimes in a different direction. In most cases errors from both inversions ranges between 2.5% and 55% except for release #12 and #32 where error reached 65% and 63% for simulated gradients assimilating the reference data respectively and release #12 with an error of 85% for simulated gradients assimilating the TGS data. The overall inversion performance assimilating TGS data and reference data are good and consistent. The slightly better average performance in the release rate

estimates using TGS data (25% error) than the estimates using reference data (30% error) is not significant in regards of the variability of the results. It highlights the weight of the model errors associated to the simulation of the average mole fraction with a Gaussian model. The results demonstrate that the errors in the release rate and location estimations from inversions using both reference and TGS data are dominated by these model errors. The errors in the reconstruction of $CH_4$ spikes from the TGS data are thus sufficiently low for use in the inverse modelling problem analysed here. One should note that, as mentioned

on section 2.7, in this study, the reference inversions rely on a restrained subset of the reference data that match the available





data from TGS sensors. Results from Kumar et al. (2022), considering the entire dataset available on the reference instruments, yielded significantly improved results.

While our work presents promising results regarding the use of low-cost MOS sensors for estimating $CH_4$ emission rates and locations, it's imperative to acknowledge the high degree of uncertainty associated with continuous emission monitoring

(CM) solutions, as evidenced by the study conducted by Bell et al. (2023). In their study, various CM technologies were tested against a series of controlled releases, revealing a broad range of true positive rates, false positive rates, and significant errors in the estimation of emission rates. Bell et al. found considerable variability in the performance of CM technologies, with mean relative errors (MRE) ranging from -44% to +586% for release rates of 0.1 - 1 kg/h and for release rates above 1 kg/h, an MRE between -40% and +93%. These findings underscore the current limitations and inconsistent performance of

CM solutions, even under less complex conditions than typically encountered in the field. While our study is encouraging, it represents just one step in the progression of this approach. Further research, rigorous testing, and critical interpretation of results are necessary for future advancements.

## 5   Conclusions

This study presents different techniques to reconstruct $CH_4$ mole fractions from the voltage signal measured by metal oxide

low cost TGS sensors deployed downwind an area of points of controlled releases during a campaign at TADI in 2019. The data from this reconstruction are assimilated in an inverse modelling framework to quantify the rate (ranging from 0.4 to 5 g $CH_4$ s$^{-1}$) and location of these controlled release. The approach employed to extract the baseline signal on TGS voltage measurements based on surrounding observations allowed us to derive and successfully correct the baseline signal on TGS sensors without the need of using other environmental parameters. The reconstruction of $CH_4$ mole fraction from voltage

observations measured during controlled releases showed good agreement with observed $CH_4$ mole fractions from the reference instruments. The reconstruction was consistently better with TGS 2611-C00 sensor regardless the reconstruction model used. Both models had met our requirement target of NRMSE of reconstructed $CH_4$ lower than 0.15 ppm across all chambers when trained with the TGS 2611-C00 sensor. Emission rate and source location estimates using an inversion based on a gaussian plume model produced similar results using reconstructed $CH_4$ mole fraction from TGS sensors data to those obtained with

high precision instruments, with an average estimate rate error of 25.5% and a mean source location error of 9.5 m from TGS data. In this study, the reconstruction of the $CH_4$ mole fractions was conducted independently from the inversion modelling. The estimation error could probably be reduced with a better understanding of inverse modelling sensitivity to the misfits from the reconstruction models. In consequence, a sensitivity study is encouraged to determine the best approach for the reconstruction of the observations from TGS sensors.

*Data availability.*  The dataset was collected in the frame of the Chaire Indutrielle TRACE ANR-17-CHIN-0004-01. It is publicly accessible at this link: https://doi.org/10.5281/zenodo.8399829



*Author contributions.* Olivier Laurent and Ford Cropley designed the Figaro® logger system. Olivier Laurent, Christopher Caldow and Ford Cropley conducted the field measurement campaign. Rodrigo Rivera and Diego Santaren developed the $CH_4$ reconstruction models. Rodrigo Rivera, Olivier Laurent and Cécile Mallet developed the baseline correction methodology of TGS sensors. Pramod Kumar and
Rodrigo Rivera developed the inversion framework to estimate the release locations and emission rates. Rodrigo Rivera, Gregoire Broquet and Philippe Ciais prepared the manuscript with collaboration of the other co-authors.

*Competing interests.* The authors declare that they have no conflict of interest.

*Acknowledgements.* This work was supported by the Chaire Industrielle Trace ANR-17-CHIN-0004-01 co-funded by the ANR French national research agency, Total Energies-Raffinage Chimie, SUEZ - Smart & Environmental Solutions and THALES ALENIA SPACE.



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



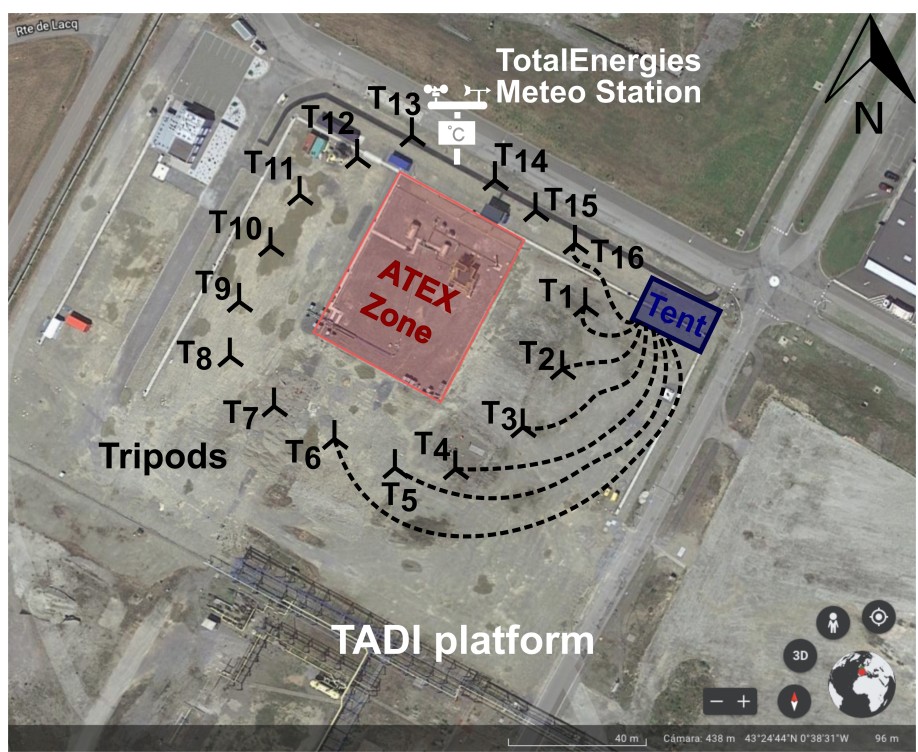

**Figure 1.** Diagram of the experimental setup on top of a satellite image of the TADI platform (source: © Google Earth). The locations of the releases are inside the red rectangle (ATEX zone). The locations of the 16 tripods are presented as black symbols and denoted with a Tx where x is the index of the tripod from 1 to 16. The blue rectangle indicates the tent location. Examples of the sampling lines connecting the tripods to the tent are shown as dashed lines, only showing 7 of 16 in total. The white symbol shows the location of the Meteorological station installed by TotalEnergies® .





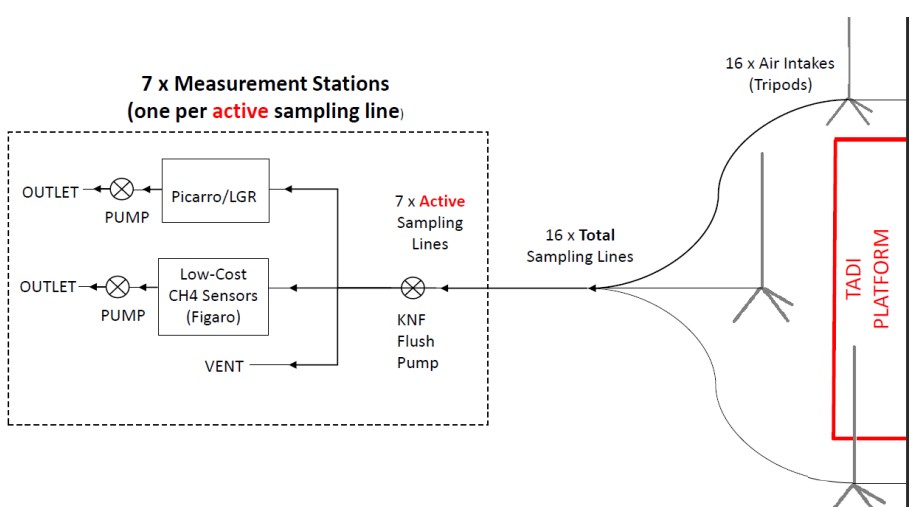

**Figure 2.** Diagram of the measurement stations and their connection to the sampling lines.



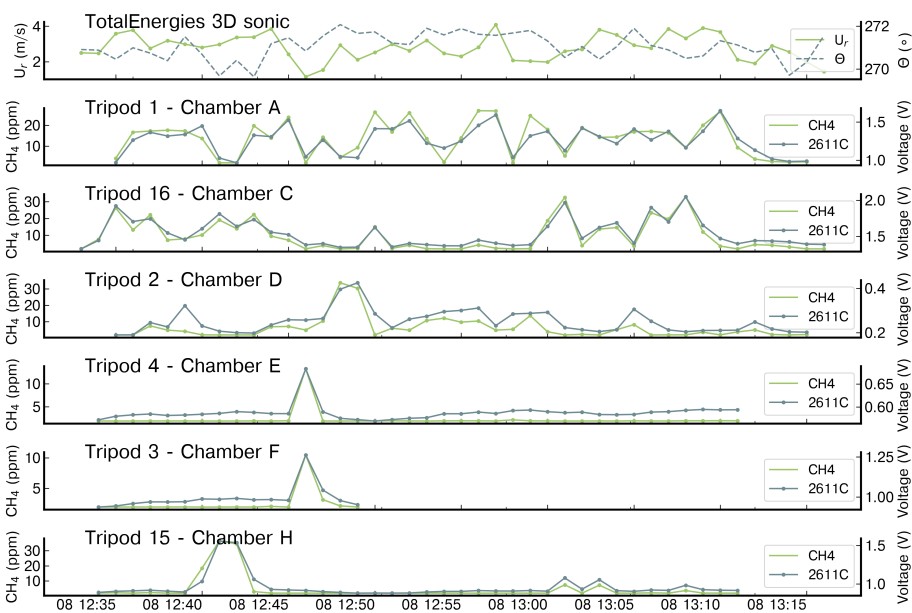

**Figure 3.** An example of 1-minute averaged $CH_4$ mole fraction (ppm) and voltage (V) measurements respectively measured by six high precision instruments and one type of TGS sensor (TGS 2611-C00) for release 25 ($Q_s = 5$ g s$^{-1}$). $CH_4$ measurements from the high precision instruments are denoted as 'CH4' and the voltage measurements from TGS sensor are denoted as '2611C'. The top panel shows the 1-minute averaged wind speed ($Ur$) and wind direction ($\theta$) measured by the 3D sonic anemometer.





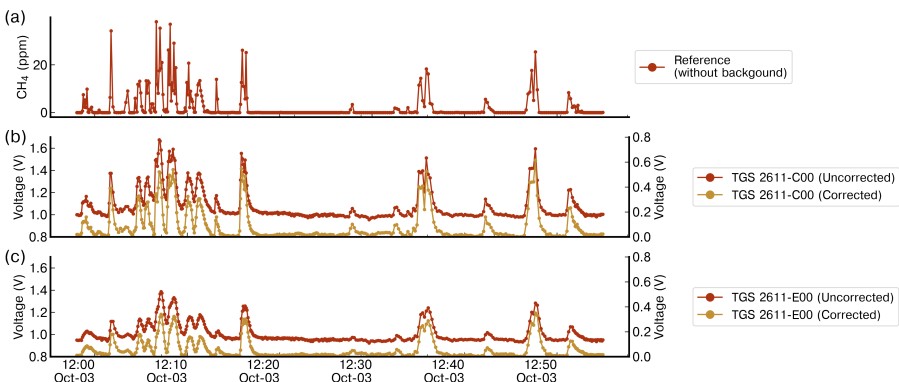

**Figure 4.** Comparison of the voltage signal for one release (#8) from Chamber A before (Uncorrected) and after (Corrected) the baseline correction on (b) TGS 2611-C00 and (c) TGS 2611-E00, on which it is appreciated the correction of the offset preserving the amplitude enhancements linked to $CH_4$ variations. (a) Reference $CH_4$ mole fractions, also corrected using the spike correction algorithm.





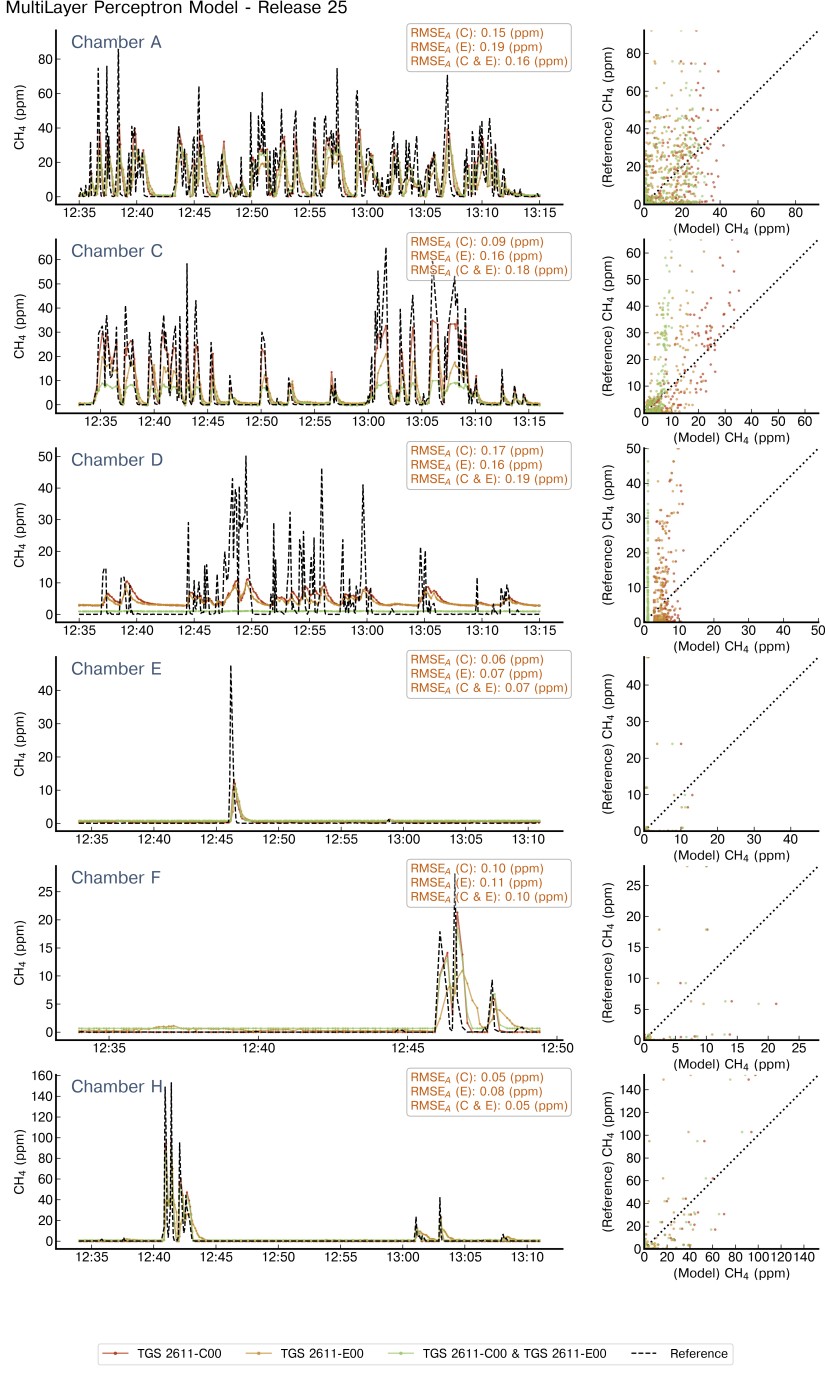

**Figure 5.** Example of reconstruction of release #25 using an MLP model. On left panels are shown the reconstructed $CH_4$ mole fractions for each chamber that captured the release, we present the reference signal (black dotted line), the reconstructed $CH_4$ mole fractions when the model has as input the TGS 2611-C00 sensor (red), the TGS 2611-E00 (yellow) or both types at the same time (green). The right panels show the 1:1 plot of the reference against the output of the model for the three configurations of inputs. Note the difference in the x-axis for Chamber F.





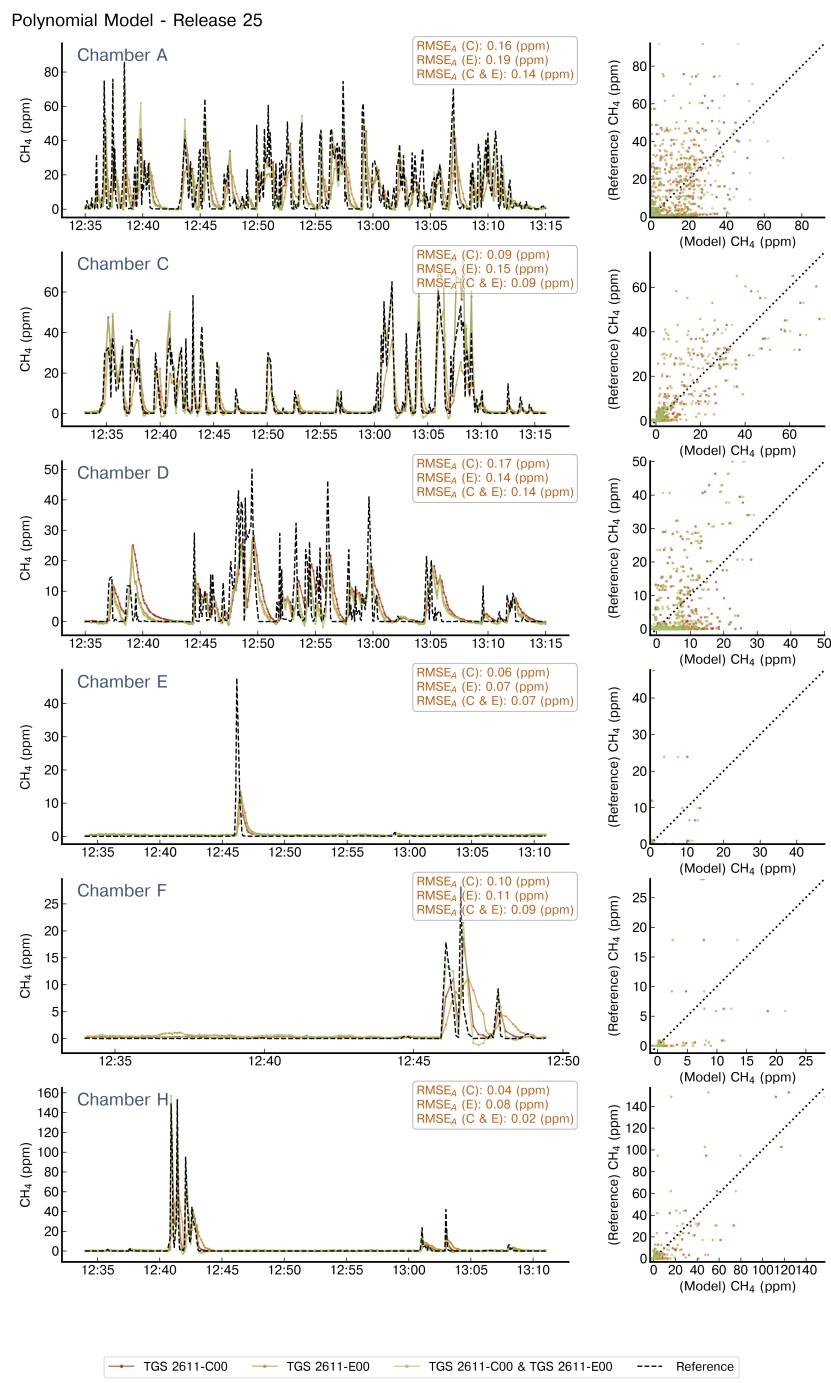

**Figure 6.** Example of reconstruction of release #25 using a Polynomial model. Notations are the same as in Figure 5.





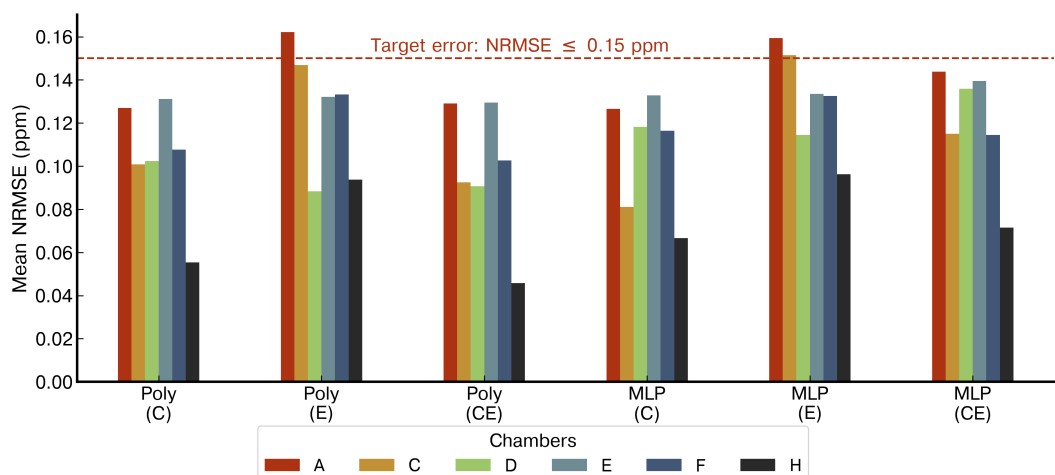

**Figure 7.** Comparison of the mean NRMSE of the two types of models trained with the three configurations of the inputs. The 2nd degree polynomials are denoted as 'Poly' and the multilayer perceptron as 'MLP'. The three input configurations are denoted inside parentheses, 'C' when the model's input was only the TGS 2611-C00, 'E' for the TGS 2611-E00 and 'CE' when both sensors were used as inputs at the same time. The color code of the bars corresponds to the chambers.





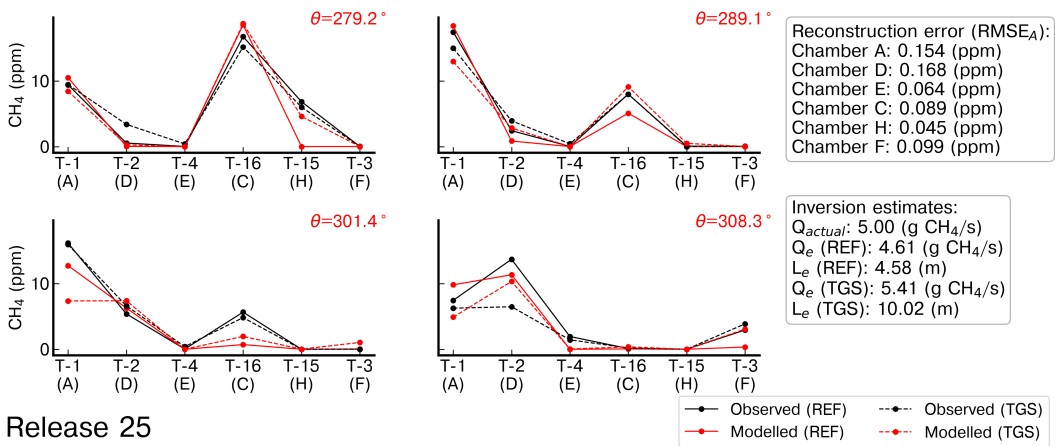

**Figure 8.** Observed and modelled average $CH_4$ mole fractions from the reference, denoted 'REF', and low-cost sensor, denoted TGS, corresponding to the release #25. The reconstructed $CH_4$ was computed using the MLP model. The index of the tripods is denoted as T-x and the average wind direction ($\theta$) for the binning of wind sectors is shown on the top right of each panel in red.



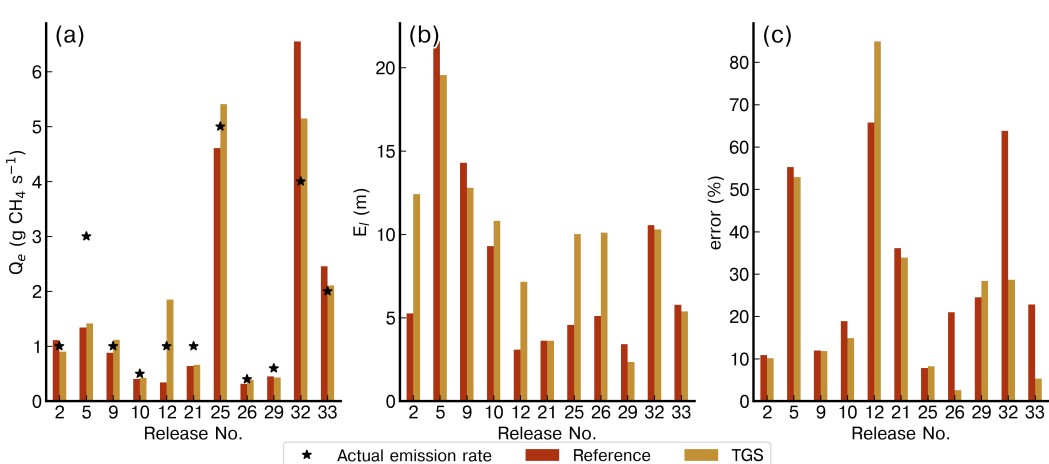

**Figure 9.** Comparison of the emission rate estimate ($Q_e$) (a), of the location error ($E_l$) (b) and of the relative error in the emission rate estimate (c) from the inversions assimilating the Reference data (in red) and the reconstruction of the $CH_4$ mole fraction from the TGS sensors (in orange). The reconstructed $CH_4$ mole fractions used in these inversions are computed with the MLP model

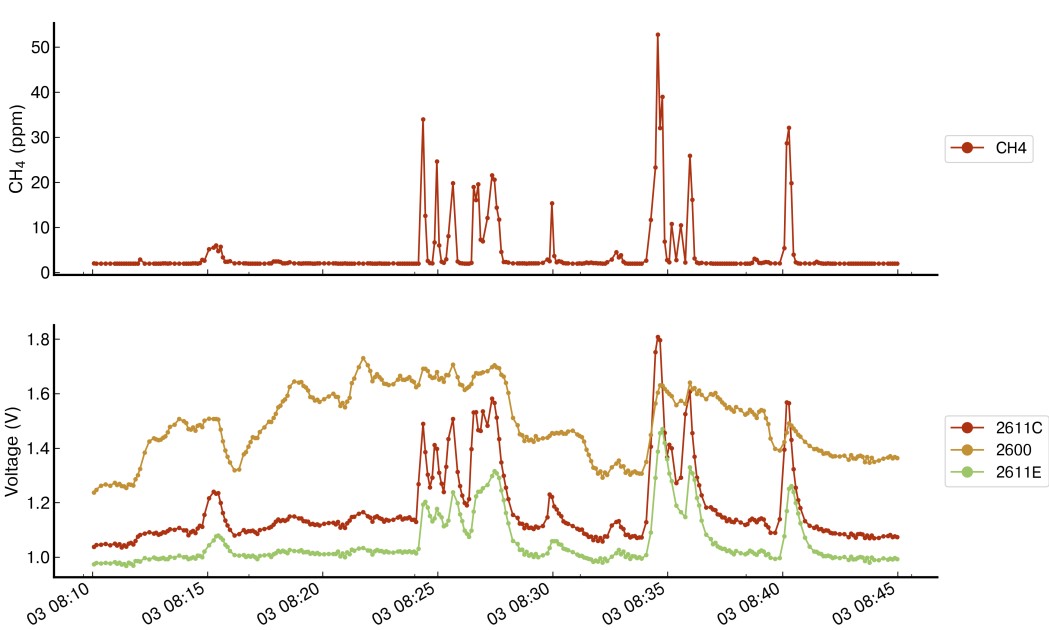

**Figure A1.** Comparison of the voltage measurements from three types of TGS included on chamber A. Upper plot shows the reference CH$_4$ observations measured from the reference instrument. Lower plot shows the voltage observations from TGS 2611-C00, 2600 and 2611-E00.



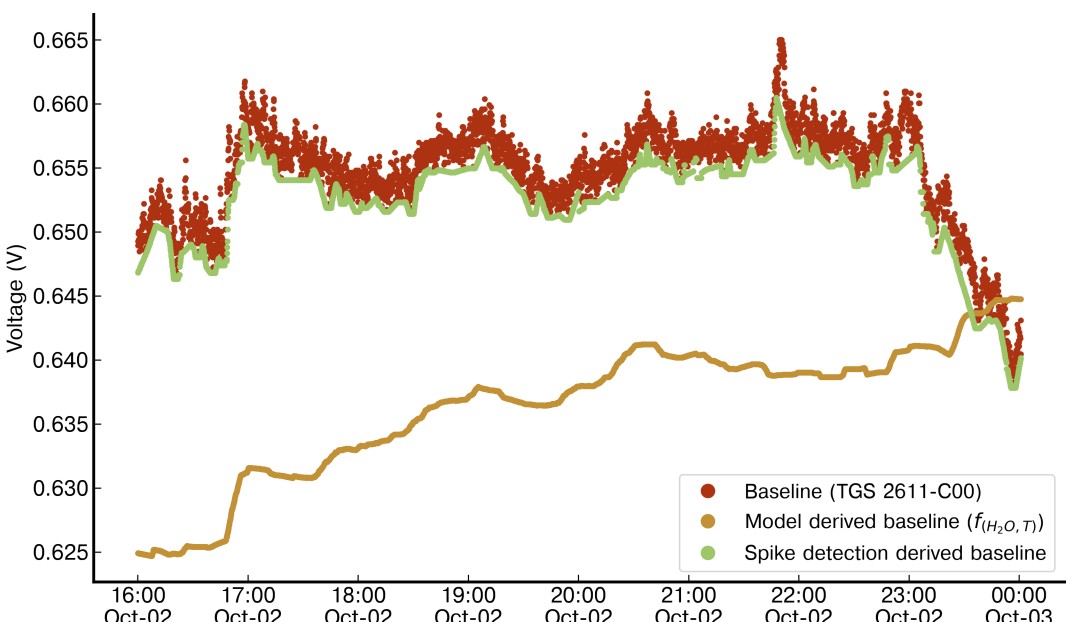

**Figure A2.** Comparison of the performance in deriving a baseline signal for the TGS 2611-C00 (red) of Chamber E between a function of $H_2O$ and Temperature (yellow) and a spike detection algorithm (green). The multilinear model derived baseline was trained on six hours of non-release periods at the start of the first day of the campaign and evaluated on the last eight hours of the same day (shown in the figure). The Spike detection algorithm, an iterative function, does not need any prior training and detects the baseline based on neighboring observations and fixed parameters.





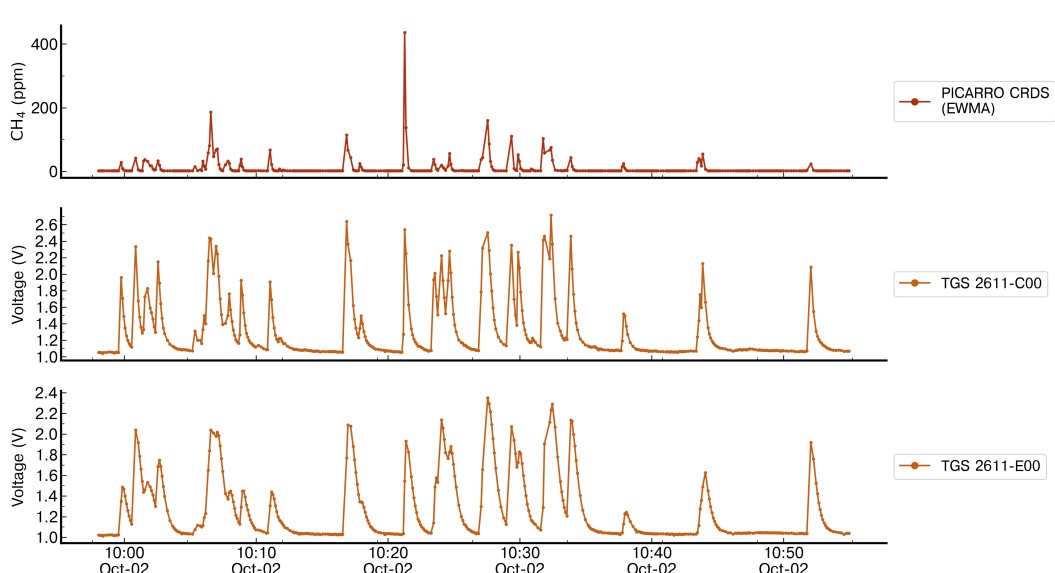

**Figure A3.** Comparison of the response of the TGS 2611-C00 and TGS 2611-E00 sensors with $CH_4$ measurements from the reference instrument for the release #2 which contains spikes with high concentration. The spikes observed on the TGS sensors corresponding from amplitudes between 100 ppm to more than 200 ppm are not distinguishable from spikes with amplitudes lower than 50 ppm.

**Figure A4.** Reconstruction of release #2 using a MLP model. On left panels are shown the reconstructed CH$_4$ mole fractions for each chamber that captured the release, we present the reference signal (black dotted line), the reconstructed CH$_4$ mole fractions when the model has as input the TGS 2611-C00 sensor (red), the TGS 2611-E00 (yellow) or both types at the same time (green). The right panels show the 1:1 plot of the reference against the output of the model for the three configurations of inputs. Note the difference in the x-axis for the chambers.





**Figure A5.** Reconstruction of release #9 using a MLP model. Notations are the same as in Figure A4. Note the difference in the x-axis for the chambers.



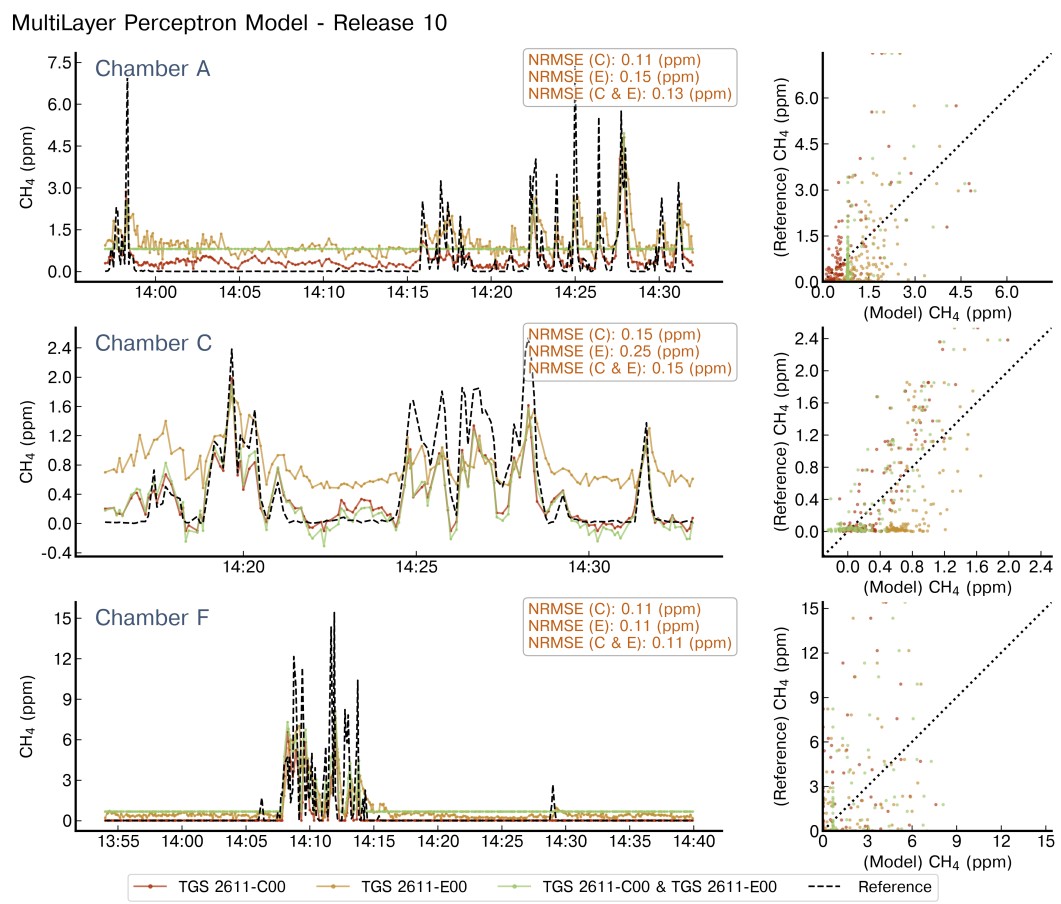

**Figure A6.** Reconstruction of release #10 using a MLP model. Notations are the same as in Figure A4. Note the difference in the x-axis for the chambers.



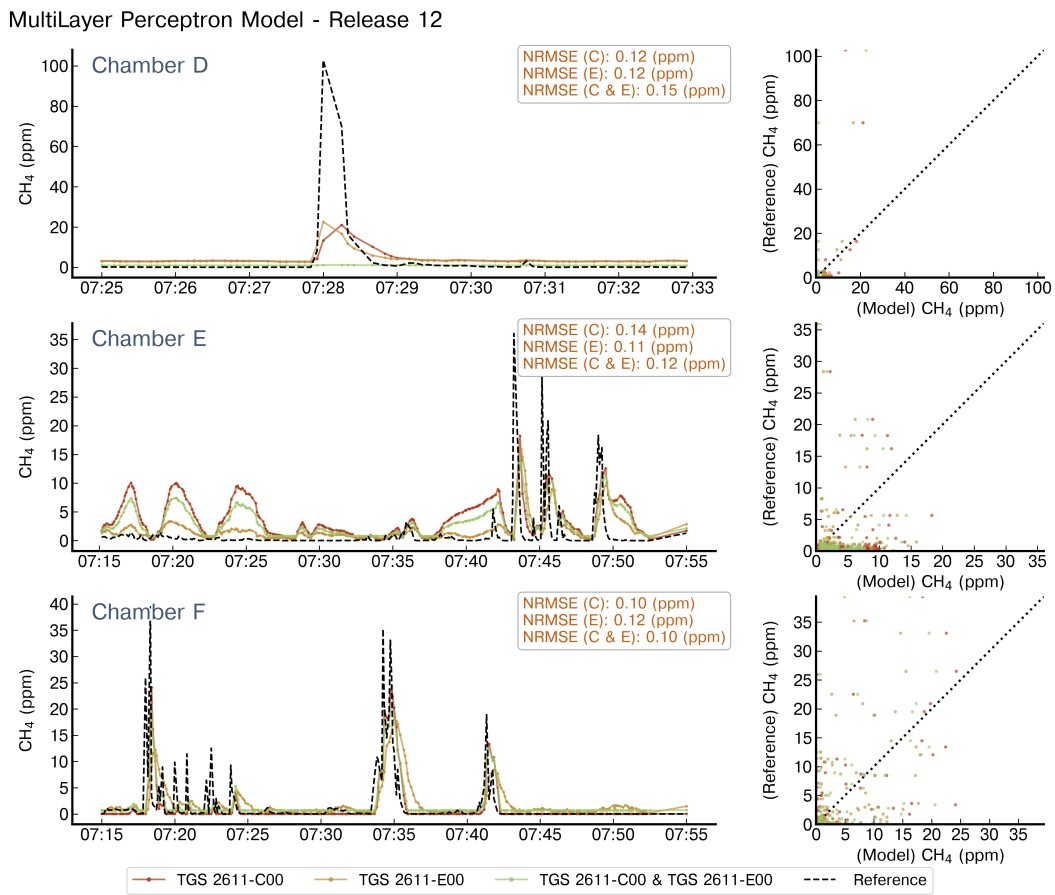

**Figure A7.** Reconstruction of release #12 using a MLP model. Notations are the same as in Figure A4. Note the difference in the x-axis for the chambers.



**Figure A8.** Reconstruction of release #21 using a MLP model. Notations are the same as in Figure A4. Note the difference in the x-axis for the chambers.





**Figure A9.** Reconstruction of release #26 using a MLP model. Notations are the same as in Figure A4. Note the difference in the x-axis for the chambers.





**Figure A10.** Reconstruction of release #9 using 2nd degree polynomials. Notations are the same as in Figure A4. Note the difference in the x-axis for the chambers.

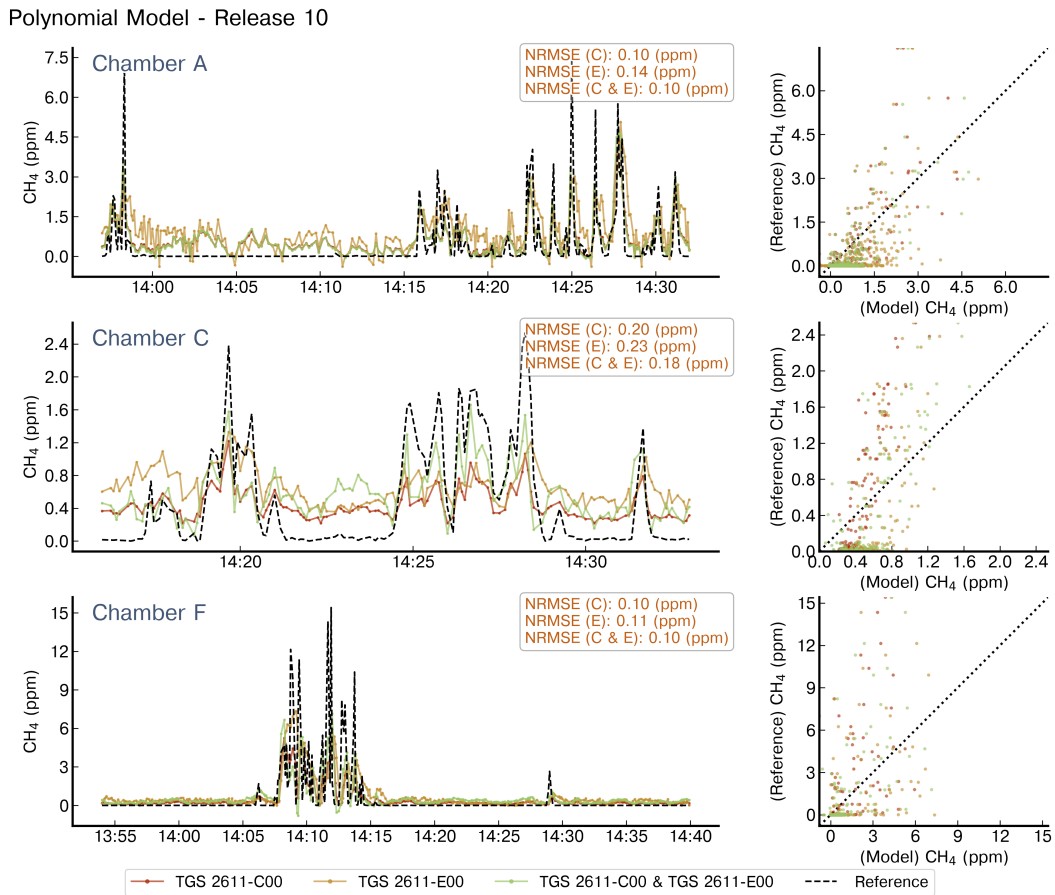

**Figure A11.** Reconstruction of release #10 using 2nd degree polynomials. Notations are the same as in Figure A4. Note the difference in the x-axis for the chambers.





**Figure A12.** Reconstruction of release #26 using 2nd degree polynomials. Notations are the same as in Figure A4. Note the difference in the x-axis for the chambers.





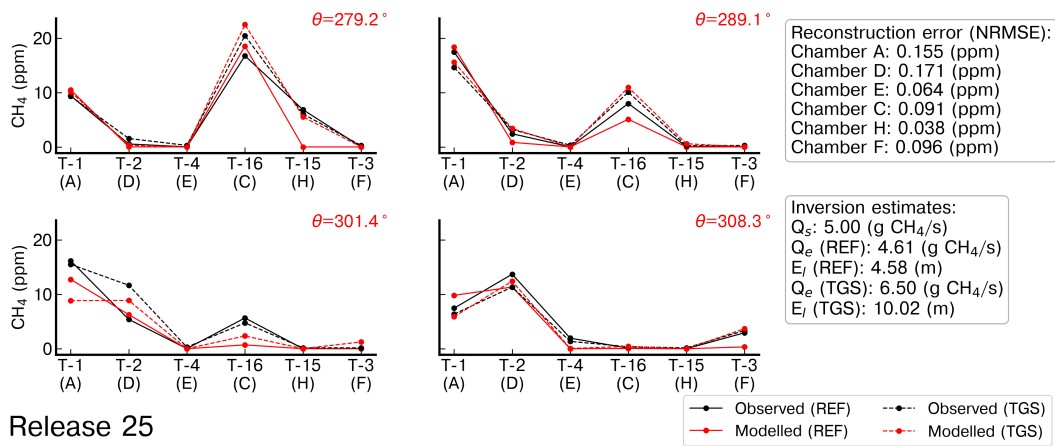

**Figure A13.** Observed and modelled average $CH_4$ mole fractions from the reference, denoted 'REF', and low-cost sensor, denoted TGS, corresponding to the release #25. The reconstructed $CH_4$ was computed using the polynomial model of 2nd degree. The index of the tripods is denoted as T-x and the average wind direction ($\theta$) for the binning of wind sectors is shown on the top right of each panel in red.





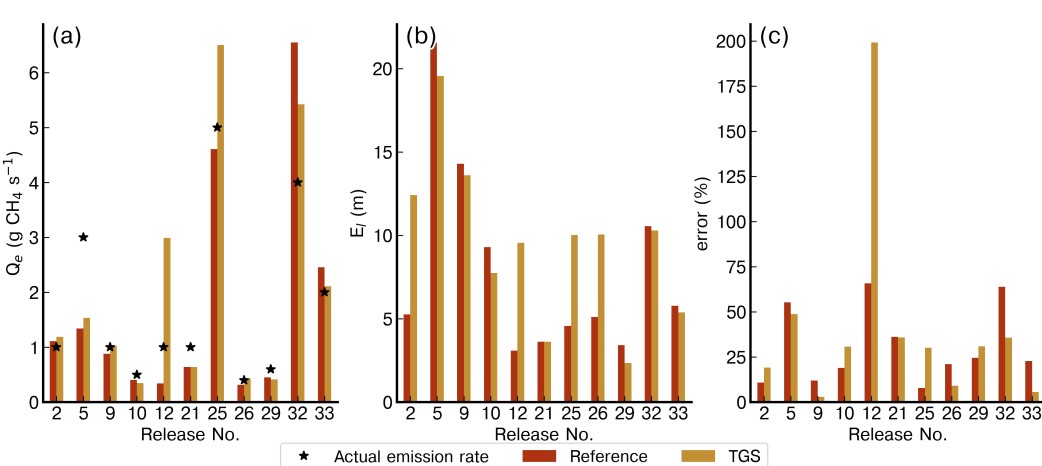

**Figure A14.** Comparison of the emission rate estimates ($Qe$), location error ($El$) and relative error on the rate estimates for the inversions assimilating the reference data and the reconstruction of the $CH_4$ from the TGS low-cost sensor based on the Polynomial model of 2nd degree.



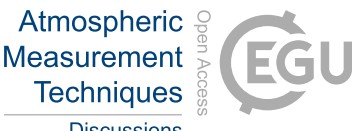

**Table A1.** Distribution of the releases by chamber. For each chamber is denoted with an 'o' the releases for which the TGS sensors produced valid measurements and with an 'x' the invalid ones.

| Release number | Chamber | | | | | | Release number | Chamber | | | | | |
|---|---|---|---|---|---|---|---|---|---|---|---|---|---|
| | A | C | D | E | F | H | | A | C | D | E | F | H |
| 1 | - | - | - | x | x | o | 19 | o | o | - | o | o | o |
| 2 | o | - | - | o | o | o | 20 | o | o | - | o | o | o |
| 3 | - | - | - | o | o | o | 21 | o | o | - | o | o | o |
| 4 | - | - | - | o | o | o | 22 | - | o | - | o | o | - |
| 5 | o | - | o | o | o | - | 23 | - | - | - | o | - | o |
| 6 | o | - | o | o | o | - | 24 | o | o | x | - | o | o |
| 7 | o | - | o | o | o | - | 25 | o | o | o | o | o | o |
| 8 | o | - | o | o | o | - | 26 | o | o | x | o | - | o |
| 9 | o | o | o | o | o | - | 27 | o | o | - | - | - | x |
| 10 | o | o | x | x | o | - | 28 | o | o | - | - | - | o |
| 11 | o | x | - | - | - | - | 29 | o | o | - | - | o | o |
| 12 | x | x | o | o | o | - | 30 | o | o | - | - | - | o |
| 13 | - | x | o | o | o | o | 31 | - | o | - | - | - | - |
| 14 | o | o | o | o | o | o | 32 | o | o | - | o | - | o |
| 15 | o | o | - | - | o | - | 33 | x | o | - | o | - | o |
| 16 | o | - | - | x | - | - | | | | | | | |
| 17 | o | o | - | - | - | x | | | | | | | |
| 18 | o | o | - | - | o | o | | | | | | | |





**Table A2.** Comparison of the emission rate estimates ($Q_s$), location error ($L_e$) and percentage of error of the rate estimates ($Q_e$) for the reference instrument and the TGS low-cost sensor from reconstructed $CH_4$ of the 2nd degree polynomial model.

| Release N° | Actual emission (g $CH_4$ s$^{-1}$) | Reference | | | TGS | | |
|---|---|---|---|---|---|---|---|
| | | $Q_e$ (g $CH_4$ s$^{-1}$) | $E_l$ (m) | error (%) | $Q_e$ (g $CH_4$ s$^{-1}$) | $E_l$ (m) | error (%) |
| 2 | 1.0 | 1.10 | 5.26 | 10.8 | 1.19 | 12,40 | 19,1 |
| 5 | 3.0 | 1.34 | 21.57 | 55.2 | 1.53 | 19,55 | 48,8 |
| 9 | 1.0 | 0.88 | 14.29 | 11.9 | 1.03 | 13,60 | 2,9 |
| 10 | 0.5 | 0.40 | 9.29 | 18.9 | 0.34 | 7,74 | 30,7 |
| 12 | 1.0 | 0.34 | 3.08 | 65.7 | 2.99 | 9,55 | 199,2 |
| 21 | 1.0 | 0.63 | 3.61 | 36.1 | 0.64 | 3,61 | 35,7 |
| 25 | 5.0 | 4.61 | 4.57 | 7.8 | 6.50 | 10,02 | 30,0 |
| 26 | 0.4 | 0.31 | 5.10 | 20.9 | 0.43 | 10,04 | 9,1 |
| 29 | 0.6 | 0.45 | 3.40 | 24.5 | 0.41 | 2,34 | 30,8 |
| 32 | 4.0 | 6.55 | 10.55 | 63.8 | 5.42 | 10,28 | 35,6 |
| 33 | 2.0 | 2.45 | 5.77 | 22.7 | 2.11 | 5,37 | 5,5 |
| **Average error** | | | 7.86 | 30.7 | | 9.5 | 40.6 |
| $\sigma_{error}$ | | | 5.46 | 20.3 | | 4.6 | 51.9 |



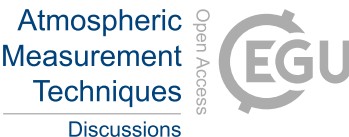

**Table A3.** Summary of the tripods that were connected to each chamber.

| Chamber | Tripod N° |
|---------|-----------|
| A | 1, 4, 6, 8, 9, 10, 11, 14, 15 |
| C | 2, 7, 9, 14, 15, 16 |
| D | 2, 3, 9, 10, 11, 12, 13, 16 |
| E | 1, 3, 4, 5, 10, 11, 12, 13, 16 |
| F | 2, 3, 4, 10, 11, 12, 13, 14, 15 |
| H | 4, 5, 6, 7, 12, 13, 14, 15 |



**Table A4.** Comparison between TGS sensors included on the low-cost logging systems during the TADI 2019 campaign.

| Type | Target gas | Approximate price | Comments |
| --- | --- | --- | --- |
| 2600 | $C_2H_5OH$, $C_4H_{10}$, $CO$, $H_2$, $CH_4$ | 15 \$us | Designed as a smoke detector. |
| 2611-C00 | $CH_4$, $C_2H_5OH$, $C_4H_{10}$, $CO$, $H_2$ | 20 \$us | Designed for $CH_4$ detection. Fast response. |
| 2611-E00 | $CH_4$, $H_2$ | 20 \$us | Designed for $CH_4$ detection. Increased selectivity due to a carbon filter installed on top of the sensing material. |
| 2602 | $C_7H_8$, $H_2S$, $C_2H_5OH$, $NH_3$, $H_2$ | 17 \$us | High sensitive to VOC and odor gases. |