# Peer review of "Using metal oxide gas sensors to estimate the emission rates and locations of methane leaks in an industrial site: assessment with controlled methane releases"

_Atmospheric Measurement Techniques, 2023_

## Referee Comment (RC2)

General comments:

This paper gives a good test on low cost sensors in detecting CH4 leakages. Using reference instruments, statistics models and atmospheric inversions, this study obtained relatively good results on methane concentrations and then the emissions rates, showing the promising use of such sensors. I think this study falls in the scope of AMT, but the MS needs further substantial revisions for potential publication in this journal.

Major comments:

1. Line106-107 showed that inversion errors from high precision measurements are 23-30% and 8-10m, and inversions from low-cost sensors can reach the same level in the abstract (25%, 9.5m), why is this? And the authors need to point these out in the abstract, which are associated with L336-337 and L410-412: "highlighting the higher impact of the model error on the inversion than the reconstruction error of CH4 mole fractions". Without high precision instruments (e.g. the background information), can this be achieved? Add the role of high precision instruments in the abstract. Since performances of inversions are associated with wind conditions, the applications are also have

such limitations, which should be pointed out.

2. Add limitations and conditions of this method and implications of this study in the abstract;

3. Line 140-141 reported that 2600 are useless, but there are reports that they are useful e.g. in Eugster et al., 2020 (AMT) , and it needs more discussions on Rs/R0 ratio, which is sensitive to methane (10-100ppm) from 0.7-1.0 in the datasheet (see below figure), and also $R_L$;

[Figure]

4. The writing and expression need substantial improvements. And many parts are very hard to follow. The manuscript needs to be polished by an experienced language editor, to thoroughly improve the fluency and remove grammar errors;

5. Discuss why E00 is bad compared with C00, e.g. in Fig.4 and 7;

6. Add designs, and photos on low-cost sensor instrument;

7. I suggest the authors provide spatial distributions of simulations and inversions for typical cases, e.g. to show the real emission sources and the inversed sources and their distances.

8. I recommend the authors to make the inversion code publicly available to improve the wide influences and applications of this study.

Minor comments:

Add regression coefficients (slope, intercept and p value) in all related figures (e.g. Fig.5-6; Fig. A4-A12) that are statistically significant.

Figure 4: Add scatter plots (and coefficients) of the corrected and reference data.

Line99: participate in;

Line100-101: ambiguous for "for the estimation of ⋯ based on ⋯high precision", better to separate this for another sentence? "And the TRACE program is ⋯";

L112: consist of doing is better to be changed to consist of sth.

L115: You may mean "connected **to an upstream chamber which holds** the high precision instruments⋯"

L116:  (Picarro  or LGR), or provide specific type;

L122: better to use the datasheet parameter: "less than 3 ppb per

month"

L124-127: hard to follow, needs to be rewritten in short sentences;

L129: redundant, combine sensors: "the $CH_4$ and environmental sensors"

L131: two  other sensors;

L134-135: add "**a**" ···ADC board ;··· change "recorded" to records

L146: Why they are used in the training of models?

L217: We use**d**

L219 and 222: presente**d**

L220: the unit of hmax is ppm? And thus the NRMSE is dimensionless?

L244: change "are" to "were"

L245: **T**able 4

L275-276: how long is the typical time decay?

L316: discuss a bit on why

L333-335: redundant and a bit ambiguous

L340, 348: comply with the journal requirements on capitals of figures and keep consistency through the text (Figure A14 and fig 9a).

L366-372: These contents seems to be more suitable for conclusion

L431-433: The study of how many sensors are needed and the layout of these sensors are also needed.

---

## Author Response (AR1)

**Author's response**

**Comments from Referees**

**Author's response**

**Author's changes in manuscript**
* * *
**RC1: The manuscript is poorly written and difficult to read in some parts (for example, the title is quite odd).**

We have improved the writing of this manuscript as highlighted below. In addition we have modified the title. The new title is:

*"Using metal oxide gas sensors to estimate the emission rates and locations of methane leaks in an industrial site: assessment with controlled methane releases"*

**RC1: The concept isn't very novel and several studies have presented the conversion of voltages output by Figaro MOS sensors into mixing ratios.**

Indeed, the conversion of these voltages into mixing ratios is not the novelty of this publication. The specific conversion used here is adapted from Rivera et al. (2023).

**RC1: Novelty is the methods by which the emission location and the emission rate are calculated.**

The atmospheric dispersion inverse modelling method to derive emission locations and rates is not the novelty of this publication either: it has been documented in detail in the publication Kumar et al. (2022).

**RC1: Details on both processes are lacking and it is currently unclear how novel the approaches used here are.**

Our recent publications explicitly refer to Rivera et al. (2023) and Kumar et al. (2022) for more details on the overall rationale, strategies, concepts, and implementations of these two processes. We have modified the text to state it more clearly.

The two major novelties of this new publication are

1) the application of the approach from Rivera et al. (2023) to on-site measurements during a real blind test of controlled releases with sub-optimal constraints that we did not manage

2) a) the combination of the two processes (derivation of mixing ratios from Figaro voltages, and atmospheric dispersion inverse modelling to derive emission location and rates from these mixing ratios) b) with a key underlying question: is the accuracy of the mixing ratio derivation satisfying for the purpose of the atmospheric dispersion inverse modelling? In previous publications, we had to derive notional targets for the accuracy/precision of the mixing ratio derivation, but the actual signal exploited by the atmospheric inversions from the timeseries of mixing ratios (through a complex binning of the mixing ratio data in practice, and then through the minimization of the model-observation misfits balanced by the modelling errors in the inversion process), and thus the requirements in terms of mixing ratio precision/accuracy are difficult to anticipate.

RC1: Also, this work would benefit from a deeper dive into the applicability of the approach in the real world.

Similar comments were already raised during the preliminary review of this manuscript, and some text had already been added in the introduction to clarify these points. We have the feeling that it is sufficient.

RC1: **Abstract: Language needs to be improved throughout. There are no details of the tests here: flow rates, meteorological conditions, time of year, how long the test were run. Too much extraneous detail between lines 7 and 14. Should be condensed. Last sentence (L14 to 16) doesn't make sense.**

We have updated the abstract to address the points highlighted by the reviewer. Here is the new version of the abstract:

Fugitive methane ($CH_4$) emissions occur in the whole chain of oil and gas production, including from extraction, transportation, storage, and distribution. Such emissions are usually detected and quantified by conducting surveys as close as possible to the source location. However, these surveys are labour intensive, are costly, and fail to not provide continuous emissions monitoring. The deployment of permanent sensor networks in the vicinity of industrial $CH_4$ emitting facilities would overcome the limitations of surveys by providing accurate emission estimates, thanks to continuous sampling of emission plumes. Yet high precision instruments are too costly to deploy in such networks. Low-cost sensors using a metal oxide semiconductor (MOS) are presented as a cheap alternative for such deployments due to their compact dimensions and to their sensitivity to $CH_4$. In this study, we demonstrate the ability of two types of MOS sensors (TGS 2611-C00 and TGS 2611-E00) manufactured by Figaro®  to reconstruct a $CH_4$ signal, as measured by a high-precision reference gas analyser, during a 7-day controlled release campaign conducted by TotalEnergies in autumn 2019 near Pau, France. We propose a baseline voltage correction linked to atmospheric $CH_4$ background variations per instrument based on an iterative comparison of neighbouring observations, i.e. data points. Two $CH_4$ mole fraction reconstruction models were compared: multilayer perceptron (MLP) and $2^{nd}$ degree polynomial. Emission estimates were then computed using an inversion approach based on the adjoint of a Gaussian dispersion model. Despite obtaining emission estimates comparable with those obtained using high precision instruments (average  emission rate error of 25% and average location error of 9.5 m), the application of these emission estimates is limited to adequate environmental conditions. Emission estimates are also influenced by model errors in the inversion process.

RC1: **Introduction: Throughout the introduction - It reads like a list of barely related bullet points. This needs to be rewritten and resubmitted for review. Currently, I find this very hard to comment on as there are too many inaccuracies and poorly written sentences. I have picked out some examples but this list is not exhaustive. For example:**

**At L18: "Fossil fuel anthropogenic methane (CH4) emissions related to the production, exploitation and transport of coal, oil and natural gas, account for 35% of global anthropogenic emissions (Saunois et al., 2020)." Isn't all incorrect, only poorly written. It would be better written as: A recent study suggests that in 20XX methane (CH4) emissions from the production, transportation, storage and distribution of fossil fuels (e.g. coal, oil and natural) account for 35% of global anthropogenic CH4 emissions (Saunois et al., 2020). Then delete the next sentence.**

We think that this is not an obvious example of a poorly written sentence. Sharing the statement from a publication and connecting this statement to the publication via simple parenthesis is very common in scientific publications to lighten the text, to avoid long series of "According to publication1…, Publication 2 indicates that….".

However, we have accepted the correction and reformulated the phrase as suggested, and we have reviewed the whole manuscript to improve the writing.

A recent study suggests that in the decade 2008-2017, methane ($CH_4$) emissions from the production, transportation, storage, and distribution of fossil fuels (e.g. coal, oil, and natural gas) accounted for 35% of global anthropogenic $CH_4$ emissions (Saunois et al., 2020).

RC1: Introduction: L26: "Atmospheric measurements" is too vague. Please describe what is being measured. "The downwind measurements CH4 concentration in the air" (or similar).

Added a clarification for "atmospheric measurements".

Atmospheric measurements are increasingly used to detect and quantify $CH_4$ leaks from industrial facilities. These methods primarily involve measuring methane mole fraction downwind of the facility.

RC1: Introduction: L37: "limiting the impact of atmospheric transport modeling uncertainties". This is the key issue with networks of sensors.

We have rewritten the phrase:

"The use over long periods of time of dense networks of sensors deployed permanently increase the ability to identify the structures of the observed plumes, to improve the atmospheric transport modelling parametrization for the simulation of these plumes, and thus to improve the accuracy of the quantification of the leaks based on this modelling."

RC1: Introduction: L38: "Chamberland and Veeravalli (2006)" this is an older study to be referencing. Also, the main issue with quantifying fugitive emissions on production sites is differentiating the vented and combustion emissions from fugitives.

Removed the phrase and reference from the manuscript.

RC1: Introduction: L40: "dense network" another key issue with monitoring O&G productions sites. How many sensors? Typically, only four are deployed by a solution provider regardless of site size. Key question is how to optimize?

We have expanded the discussion section to cover the density of the sensor network. This discussion outlines our empirical approach in deploying a network of 16 tripods equipped with dual-sensor setups at strategic distances ranging from 40 to 50 metres from the emission source. Our findings suggest that while the industry standard might lean towards deploying only four sensors regardless of site size, our campaign demonstrates that a configuration of 4 to 5 sensors, strategically placed, can effectively minimise errors in methane emission quantification.

We acknowledge that the question of optimising sensor count and placement is critical, especially considering varying site sizes and environmental conditions. Our analysis,

particularly of cases with uniform emission rates (1 g CH4/s), indicates that a careful balance between sensor count and their strategic placement plays a pivotal role in achieving accurate emission estimations. However, we also highlight that determining the optimal configuration for a sensor network is complex and warrants a thorough investigation, potentially through simulations that explore typical emissions scenarios and the impact of adding or removing sensors. Given the scope of our study and the limited data points captured, a comprehensive exploration of optimal sensor network configuration was beyond our research boundaries.

RC1: Introduction: L43: "metal oxide sensing material" not strictly true. The metal oxide doesn't sense anything, more physically changes in the presence of methane.

Rephrased to improve clarity. The new phrase is:

MOS sensors are composed of a metal oxide sensing material, incorporating an integrated heater. A chemical reaction affects the electrical conductivity of the sensing material in the presence of an electron donor gas such as $CH_4$ (Örnek and Karlik, 2012)

RC1: Introduction: L46: "suited to long time deployment" Again, not true. These sensors require relatively high amounts of energy (as you state in the following sentence), work poorly in low humidity and work poorly in low temperatures (as you state in the nest sentence).

Accepted correction. The content was reformulated, this is the new version of the text:

However, MOS sensor sensitivity is also affected by other environmental parameters such as temperature and relative humidity (Popoola et al., 2018); they also present low accuracy and may drift with time (in the form of a decrease in the conductivity of the sensing material), requiring periodic re-calibrations (Riddick et al., 2020; Shah et al., 2023, 2024), and the need for a constant power supply due to the heater material (Shah et al., 2023).

RC1: Introduction: L62 – L71 Suggest you read: Riddick, S. N., Ancona, R., Cheptonui, F., Bell, C. S., Duggan, A., Bennett, K. E. and Zimmerle, D. J. (2022) A cautionary report of calculating methane emissions using low-cost fence-line sensors. Elementa: Science of the Anthropocene 10(1). https://doi.org/10.1525/elementa.2022.00021

The study of Riddick et al. (2022) was added to the introduction. Here is the new version of the paragraph:

The next logical step is to test the performance of the same sensors to reconstruct $CH_4$ mole fraction from real leaks, and to use reconstructed mole fractions to quantify and localise emission rates. Riddick et al. (2020) quantified $CH_4$ emissions from a gas terminal using a Figaro TGS 2600, deployed 1.5 km from the emission source. To reconstruct $CH_4$ mole fractions from voltage observations, the authors developed an empirical equation considering the measured voltage, temperature, and relative humidity from a nearby meteorological station and then applied a Gaussian plume model to quantify the emission rate using reconstructed $CH_4$ mole fractions and local wind information. Their estimated average emissions of 9.6 g $CH_4$ $s^{-1}$, with a maximum emission rate of 238 g $CH_4$ $s^{-1}$, given corresponding $CH_4$ mole fraction enhancements of between 2 ppm to 5.4 ppm within the plume. Their Figaro-based emission estimates were not confronted with corresponding emission estimates derived using a high precision gas analyser nor with an independent knowledge of the emission rate. Elsewhere,

Riddick et al. (2022) studied the capabilities to detect and estimate $CH_4$ emissions of four Figaro TGS 2611-E00 sensors in a fence-line monitoring setup. Sensors were deployed closer to the emission source (30 m) and tested over a 48 hour period. Reported results showed detection consistency for emissions above 167 g $CH_4$ $h^{-1}$ with an enhancement threshold of 2 ppm. However, the number of sensors used to compute the emission estimates was not specified, particularly given the spatial distribution of the sensors and varying wind speed.

**RC1: Introduction: L78 – L109 Much of this is methods and should be condensed.**

**The text was updated to present the context and contribution of the study more clearly.**

This study builds upon the research conducted by Rivera et al., (2022) and Kumar et al. (2022), demonstrating the potential for continuous monitoring of $CH_4$ emissions using cost-effective in situ sensors. Drawing from the insights derived from these two studies, it seeks to address the new challenges associated with the combination of both types of analysis, i.e reconstruction of $CH_4$ mole fractions from measured voltage variations and estimation of emission rates and location from $CH_4$ mole fractions. Firstly, the challenge arises in the deployment and management of onsite Figaro sensors, an issue not present in Rivera et al. (2022), as well as extracting $CH_4$ mole fractions from measurements that are impacted by more complex perturbations. For instance, the background air in Rivera et al., (2021, 2022) was less polluted than air from an industrial site such as TADI. Moreover, the environmental conditions, especially in terms of temperature and water mole fraction, in these previous studies were smooth and not representative of field conditions as encountered in this new study. Secondly, the prescriptive precision and accuracy targets for $CH_4$ reconstructions outlined in Rivera et al. (2022) were established as generic targets, fitting for a variety of data processing strategies intended to quantify emissions from industrial sites. The specific observation and modelling strategy implemented in Kumar et al., (2022) to localise and quantify point source emissions carries its own set of precision/accuracy requirements. In particular, this strategy strongly relies on the characterisation of gradients across measurement stations of mole fraction averages over time or wind sectors, which makes the derivation of nominal requirements on the reconstruction of $CH_4$ spikes or the $CH_4$ time series quite complex. Furthermore, such requirements should be weighed against the modelling uncertainties associated with the corresponding Gaussian plume model inversions. Ideally, the uncertainties related with $CH_4$ mole fraction data would not significantly contribute towards the total uncertainty when combined with uncertainties from the modelling framework. This, however, does not necessarily mean that they should be much smaller than the latter. The direct comparison of the results obtained in this study with $CH_4$ mole fraction data derived from Figaro sensors and those from Kumar et al., (2022) provides insights into whether this objective is achieved.

Therefore, for 33 controlled releases at the TADI facility, we employed fixed-point measurements from both high precision CRDS instruments and low-cost TGSs. A considerable fraction of the TGS measurements were used for training models to reconstruct $CH_4$ mole fractions from measured TGS resistance and other variables. When reconstructing $CH_4$ mole fractions, we proposed a minimum accuracy target of 15% the amplitude of the largest observed mole fraction enhancement within a release. This corresponds to accuracies from 0.3 ppm for a release causing a maximum enhancement of 2.4 ppm up to 18 ppm for a maximum enhancement of 120 ppm. This accuracy is consistent with the accuracy requirement imposed in our previous study where we used TGS sensors to reconstruct $CH_4$ spikes created in a

laboratory experiment (Rivera Martinez et al., 2022). However, the relevance of this target is implicitly re-assessed through the use of the reconstructed time series in the inversion scheme from Kumar et al. (2022).

**RC1: Methods:**

The section was reorganised to present the methods in a more coherent and logical manner. In the current methods section we first introduce the TADI 2019 campaign (Section 2.1), then we present the Controlled releases and the sampling configuration (Section 2.2). The low cost logger system and the meteorological data is presented in Section 2.3. On Section 2.4 we talk about the preprocessing steps of the TGS data. Section 2.5 describes the reconstruction of Ch4 mole fraction from TGS voltage data and the metric used to evaluate the performance of reconstruction models. The selection of the training and testing sets are presented in section 2.6. Finally, Section 2.7 presents the inverse modelling framework used to estimate the release rates and locations.

**RC1: Methods: L112 – L128:** Please condense this section to a description of the site. Please remove any extra detail superfluous to this study.

The description of the site was updated, the corrected text is shown below:

In October 2019, TotalEnergies® performed multiple controlled releases at the TotalEnergies Anomaly Detection Initiative (TADI) facility, to investigate the capability of different detection and quantification techniques of $CH_4$ emissions from industrial facilities. The TADI test site is located northwest of Pau, France, with an approximate area of 200 $m^2$. It is equipped with infrastructure typical of oil and gas facilities (pipes, valves, tanks, etc) to simulate 'realistic' leaks. The terrain is flat but includes different obstacles that can affect the dispersion of the gases released to the atmosphere. Our experiment consisted of 41 controlled releases of $CH_4$ and $CO_2$, covering a wide range of emission rates of between 0.15 and 150 g $CH_4$ $s^{-1}$, with durations ranging between 25 to 75 minutes. We participated in this experiment to develop and test inverse modelling frameworks within the TRAcking Carbon Emissions (TRACE, https://trace.lsce.ipsl.fr/) program for the estimation of emission location and rates based on $CH_4$ mole fractions from high precision instruments (Kumar et al., 2022). We presented the inversion results for 26 releases from single point sources based on two inversion approaches, one relying on fixed-point measurements, and the other one on mobile near-surface measurements (the latter had already been documented in Kumar et al. (2021)). In both cases, the emission estimates relied on $CH_4$ mole fractions from high precision instruments, and on a Gaussian plume model to simulate the local atmospheric dispersion of $CH_4$. The results from Kumar et al. (2022) for point source emissions yielded an emission rate error of between ~23 to ~30 % and a localisation error (within a 40 m × 50 m area) of between 8 and 10 m. The controlled releases were emitted from heights of between 0.1 m and 6 m above ground level, and inside the 40 m × 50 m ATEX (ATmospheres EXplosibles) zone of the TADI facility (see Fig. 1).

**RC1: Methods: L129:** Which heights? What is the ATEX zone?

The heights and the ATEX acronym meaning were added to the manuscript. Here is the new text:

The controlled releases were emitted from heights of between 0.1 m and 6 m above ground level, and inside the 40 m × 50 m ATEX (ATmospheres EXplosibles) zone of the TADI facility (see Fig. 1).

RC1: Methods: L131:   I would like the information here instead of having to find a different paper.

We acknowledge the remark and we have removed the phrase. The relevant information of the TADI test site for this study is included in section 2.1.

RC1: Methods: L136:   I have concerns over air drawn through 100m of tubing at 6 lpm.  I assume transit time will be significant as will gas mixing in the tubing.

The transit time for air through a 10 m length of 1/4" OD Synflex tubing is approximately 1.5 seconds, and for a 100 m length, it extends to approximately 14.5 seconds. We have empirically determined these transit times on-site by introducing a short exhalation of breath into the inlet tubing and subsequently measuring the time taken to detect the resultant $CO_2/H_2O$ spike. The precise timing of these spikes was captured using GPS timestamps, ensuring high accuracy in our measurements.

Regarding the potential impact of gas mixing within the tubing, with the Reynolds number around 1970. This indicates that the flow regime within the tubing remains laminar, significantly limiting the extent of mixing. Consequently, our findings suggest that gas mixing within the tubing is minimal and should not adversely affect the measurement of controlled methane releases. In addition, we have aligned our data to correct any time shift that might arise due to transit delays, thereby preserving the accuracy of our emission estimates.

RC1: Methods: Tables 1 & 2 aren't adding much.  Could go to SI

Tables were moved to SI.

RC1: Methods: L150 – L158:  How were the supply voltage controlled?  Also, sensitivity would be affected by the ADC used, can you comment on that?

The power supply to our sensor system was fixed at a constant 5V throughout the duration of the experiment.

Regarding the analog-to-digital converter (ADC) and its impact on our measurements, it is important to clarify that while the ADC plays a critical role in defining the resolution and accuracy of the voltage readings, it does not directly affect the sensitivity of the sensor itself. The sensitivity of the sensor remains determined by its inherent properties and the environmental conditions under which it operates. The ADC's function in our setup was to accurately translate the analog signals from the sensor into digital data for analysis. Any potential disparities between the measured voltage and the actual voltage due to the ADC's characteristics were consistent throughout the experiment. This consistency ensures that while the ADC's resolution and accuracy are crucial for precise data capture, they do not alter the sensor's sensitivity to methane concentrations.

**RC1:** **Methods:** Section 2.3 – Please filter this for unnecessary information. Why mention data that were not used?

The text now belongs to section 2.2. The following text that mentions the controlled releases was updated to include only information relevant to the study and explain our motivation to use only a subset of 33 releases from the original 41 conducted during the campaign:

A total of 41 controlled releases were conducted over a seven day period, between 2 October 2019 and 10 October 2019. Six releases corresponding to low wind speeds (< 0.6 m s⁻¹) were not used for the inversion as in Kumar et al. (2022), since measurements made in low winds are not suitable for atmospheric inverse modelling. They could however be used in the training of $CH_4$ mole fraction reconstruction models. The two largest releases produced high $CH_4$ mole fraction plumes that affected the amplitude measured by the TGS sensors, such that it was not possible to distinguish large $CH_4$ spikes from medium and small spikes from voltage drop measurements (see fig A3) and, hence, they were also removed. Therefore, our study is focused on 33 from the initial 41 controlled releases conducted during the campaign. Table A1 details the releases that were measured by each chamber.

**RC1:** **Methods:** L173 – Not being able to detect < 4 ppm is concerning.

The challenge of not being able to detect methane concentrations below 4 ppm arises from a combination of factors that influence the sensitivity and accuracy of our sensor system. Firstly, the presence of a strong baseline signal at the Lacq site, caused by variations in relative humidity (RH), significantly impacts the sensor's ability to detect low methane concentrations. Additionally, discrepancies in the resistances of some loggers, which used 50kOhm resistors instead of 5kOhm, could have altered the sensitivity of the voltage measurements. These factors collectively contribute to the observed limitation in detecting methane concentrations below 4 ppm on some of the releases.

**RC1:** **Methods:**L177 – Why 1-minute averages? Seems a long time. Was the sonic near anything else?

Although the 3-D sonic anemometer measured variables at high frequency (20 Hz), due to logistical constraints on field, we would only have had access to the 1-min average data. However, our atmospheric inverse modelling framework relies on the process of average winds, especially since we use a Gaussian plume model to simulate the local dispersion. Therefore, there was no incentive to ensure that we could have access to higher frequency data.

Using data collected at shorter temporal scales, we commonly compute averages of meteorological variables such as wind speed and wind direction over longer durations or in specific wind sectors. However, computations of turbulent fluxes to use in the dispersion model to parameterize the diffusion parameters generally require high-frequency data.

**RC1:** **Methods:** Table 3 -Why were these emission rates chosen? 1.4 to 18 kg h-1 are quite high emissions, what would be typical emission rates on a production site? What have other people measured as fugitives?

The emission rates were chosen by the operator, TotalEnergies, to encompass the typical range of emissions encountered in production sites. This decision was based on the objective to ensure

that our measurements and subsequent analyses would be representative of real-world conditions, as well as to provide insights into the scale of emissions that could be expected under various operational scenarios.

**RC1:** **Methods:** Section 2.4 – This is particularly badly written. I am finding it difficult to follow. It also has too much information that could be put in the SI. Essentially TGS data were fitted to the methane analyzer data.

This section, now in section 2.5, was updated to include only the relevant information to our study.

**RC1:** **Methods:** L197 – Can you described the strengths and shortcomings of the two approaches. The main issue with approach 2 is the TGS sensors give erroneous output in low humidity conditions. This could strongly affect your linear interpolation if properly considered. At the very least, I would have thought the data were filtered for low RH.

Regarding the first approach, using $H_2O$ mole fraction and T data for correction, offers direct consideration of environmental variables affecting sensor accuracy. Its strength lies in its context-sensitive adjustment, aligning sensor data closely with real-time environmental conditions. However, its primary shortcoming in our study was the inconsistent availability of $H_2O$ and T data due to logging issues, which limited its comprehensive application.

Respecting the second approach, involving linear interpolation to establish a baseline from non-peak voltages, provides a practical solution when environmental data are lacking. This method's strength is its independence from environmental data, making it applicable even with incomplete datasets. However, its shortcoming, as pointed out, is its vulnerability to erroneous outputs in low humidity conditions. TGS sensors' performance can be significantly skewed in such environments, potentially affecting the accuracy of the interpolated baseline. Implementing a filter for low RH would have significantly reduced the amount of usable data in our study, limiting the number of available releases for analysis

**RC1:** **Methods:** Section 2.6 – Again, should be condensed.

The text was rewritten to improve clarity:

[revised manuscript text omitted]

RC1: Methods: L282: What data were used in the Gaussian plume model? Please be explicit.

See the answer to the previous comment and the update of the section 2.7

RC1: Methods: How do you overcome the GP's inability to work at distances less than 100 m?

Gaussian plume models have regularly been used to model the local atmospheric dispersion over such short distances, see (Venkatram et al., 2004, Korsakissok and Mallet, 2009; Sharan and Kumar 2009, Korbeń et al. 2022, etc.). In fact, Kumar et al. (2022) evaluated the Gaussian plume model with actual source parameters for this experimental campaign at such distances and found it to be suitable for simulations of averaged plumes at such short distances. Here, we simulate average plumes for the different wind sectors during each release, which strengthens the relevance of using such a model. Furthermore, the good accuracy of our results demonstrates that the model was suitable for such an atmospheric dispersion inverse modelling problem. We have added a sentence in section 4 to highlight the use of GP models over such short distances in a series of past studies.

RC1: Methods: L290 – L293: I do not know what this means. Please rewrite.

The text was rewritten. See answer for Comment on Section 2.7.

RC1: Results: The current information presented here could be condensed.
The result section was updated improving clarity and condensing the information.

**RC1:** **Results: I would really like to know how applicable this method would be in the real world. For example, a well pad does not only have emissions from a single fugitive point which remain active until someone fixes it. Need to consider and comment on:**

- **There could be multiple emissions at the same time: vented, combustion and fugitive. How would this approach distinguish between types? Emissions (all types) can be intermittent. How long does an emission have to run to be detected?**

    Added a paragraph on the discussion section:

    The inversion approach applied here, described for a single emission source only as required by the controlled releases experiments, can be easily extended to estimate emissions from multiple sources (see Singh et al. 2012). However, estimating emissions from multiple sources may require a more denser network of sensors to constrain a larger number of parameters for all sources. However, significant uncertainties may arise in emission estimates when measurements are taken in very close proximity to the emission sources. Our methodology requires that emissions be sustained long enough to be captured within the sampling intervals. The principal limitation is the requirement for at least four 1-minute averages, restraining the detection of short-lived emissions. Another challenge lies in the detection of emission types, such as vented, combustion or fugitive emissions. This aspect, out of the scope of our study, would require a detailed study of the characteristics of each kind of emission requiring additional tools to distinguish the particularities of them.

- **Gas coming out could be very hot, how does this affect your approach? Are any of the Gaussian Plume assumption violated? How does the GP approach work with distances < 100m?**

    The elevated temperature of the gas emission is indeed a crucial factor to consider. In the Gaussian plume model, we need to consider the plume rise for a hot gas to include the effective source height in the computations. As mentioned before, the Gaussian plume model has been applied for distances less than 100 m. In fact, recently, Korbeń et al. (2022) applied the Gaussian plume model approach for the quantification of methane emission rate from oil and gas wells in Romania using ground-based measurement techniques less than 100 m. However, as mentioned previously and discussed in the revised manuscript, there may be significant modelling uncertainties for the measurements taken in close proximity to the emission source.

- **Density of sensors – is it realistic to have 16 sensor locations on a site? What is the optimum for your approach and what is realistic?**

    The following text was added to the discussion section and table 3 was updated including the number of sensors used to compute the emission estimates for each release:

**Density of sensor network**

In our campaign, we deployed 7 chambers connected to air inlets placed on tripods at distances of between 40 and 50 m from the emission source to capture methane plumes under various conditions. Table 3 details the number of sensors used for emission flux estimations across the controlled releases.

The optimal number of sensors for emission flux localisation and estimation is complex, influenced by varying emission rates, environmental conditions, and setup configurations. Notably, when examining cases with uniform emission rates (1 g CH4/s), such as releases 12, 2, and 21 (with 3, 4 and 5 chambers respectively), a configuration of 4 to 5 sensors consistently produced the lowest errors for both sensor types. Yet, release 21 demonstrated that even five sensors may not guarantee low errors if the plume capture is suboptimal due to environmental factors or sensor placement.

We can contrast our setup with Riddick et al. (2022), who used four sensors approximately 30 m away from the source, but without detailing their individual contributions to emission calculations.

The optimal configuration of such a relatively dense network necessitates a thorough investigation, possibly through simulations of typical emissions and the strategic addition or removal of sensors to assess their impact. However, a comprehensive analysis of optimal network configuration was beyond the scope of our study due to the limited number of data points recorded."

- **Computational time. How long does it take to calculate the RSS matrix? Is this realistic?**

  The following text was added to the discussion section. Table A7 was added to SI.

  **Computational efficiency of the inversion framework**

  Our inversion framework, developed in Python 3.8 utilising numpy, pandas, and scipy libraries, efficiently computes the RSS matrix through vectorized operations and a nested for-loop. This approach achieves an average computation time per release of 0.1 seconds for the RSS matrix and 1.46 seconds for full code execution, including data preprocessing on an 8-core Apple Silicon M1 processor. The framework, which can be further optimised with multiprocessing, is detailed in Table A7, showcasing computational times across different releases. It effectively estimates emission rates and source locations on a fine grid (40m x 50m x 8m, discretized at 1m x 1m x 0.5m), demonstrating practicality for real-world applications at minimal computational costs.

**RC1:** Technical comments: Currently, the manuscript is difficult to follow and I strongly suggest a comprehensive rewrite.

We have undertaken a thorough review and comprehensive revision of our manuscript to enhance its clarity and coherence. This revision includes significant improvements to the grammar and style to ensure the manuscript is accessible and easy to follow.

We thank the reviewer for his helpful review of our manuscript. We have carefully considered all the comments and revised the manuscript accordingly. We have noted that the references to specific lines of the manuscript and some remarks correspond to the initial submitted version and do not consider the updated changes to the manuscript after the preliminary review on the current version.

Below our answers to the commentaries raised by the reviewer.

**RC2:** Line106-107 showed that inversion errors from high precision measurements are 23-30% and 8-10m, and inversions from low-cost sensors can reach the same level in the abstract (25%, 9.5m), why is this? And the authors need to point these out in the abstract, which are associated with L336-337 and L410-412: "highlighting the higher impact of the model error on the inversion than the reconstruction error of CH4 mole fractions". Without high precision instruments (e.g. the background information), can this be achieved? Add the role of high precision instruments in the abstract. Since performances of inversions are associated with wind conditions, the applications also have such limitations, which should be pointed out. Add limitations and conditions of this method and implications of this study in the abstract.

We have updated the abstract considering the comments raised by the reviewer:

Fugitive methane ($CH_4$) emissions occur in the whole chain of oil and gas production, including from extraction, transportation, storage, and distribution. Such emissions are usually detected and quantified by conducting surveys as close as possible to the source location. However, these surveys are labour intensive, are costly, and fail to not provide continuous emissions monitoring. The deployment of permanent sensor networks in the vicinity of industrial $CH_4$ emitting facilities would overcome the limitations of surveys by providing accurate emission estimates, thanks to continuous sampling of emission plumes. Yet high precision instruments are too costly to deploy in such networks. Low-cost sensors using a metal oxide semiconductor (MOS) are presented as a cheap alternative for such deployments due to their compact dimensions and to their sensitivity to $CH_4$. In this study, we demonstrate the ability of two types of MOS sensors (TGS 2611-C00 and TGS 2611-E00) manufactured by Figaro® to reconstruct a $CH_4$ signal, as measured by a high-precision reference gas analyser, during a 7-day controlled release campaign conducted by TotalEnergies in autumn 2019 near Pau, France. We propose a baseline voltage correction linked to atmospheric $CH_4$ background variations per instrument based on an iterative comparison of neighbouring observations, i.e. data points. Two $CH_4$ mole fraction reconstruction models were compared: multilayer perceptron (MLP) and $2^{nd}$ degree polynomial. Emission estimates were then computed using an inversion approach based on the adjoint of a Gaussian dispersion model. Despite obtaining emission estimates comparable with those obtained using high precision instruments (average emission rate error of 25% and average location error of 9.5 m), the application of these emission estimates is limited to adequate environmental conditions. Emission estimates are also influenced by model errors in the inversion process.

**RC2:** Line 140-141 reported that 2600 are useless, but there are reports that they are useful e.g. in Eugster et al., 2020 (AMT), and it needs more discussions on Rs/R0 ratio, which is sensitive to methane (10-100ppm) from 0.7-1.0 in the datasheet (see below figure), and also RL:

**Sensitivity Characteristics:**

[Figure]

Indeed, the datasheet shows that TGS 2600 is sensitive to CH4, as other studies (Eugster et al., 2020, 2019; Riddick et al., 2022, 2020) have used it to derive CH4 mole fractions. Our motivation to exclude this sensor from the study was based on the observed response to CH4 enhancements of the controlled releases. Figure A1 shows a comparison of a typical signal measured by the three sensors for one controlled release, as well as the signal measured by the reference instrument. The TGS 2611-C00 and TGS 2611-E00 sensors show voltage drops that correlate with the signal measured by the reference instrument, whereas for TGS 2600, only low-frequency voltage variations are observed. This inability to observe such high-frequency variations of CH4 mole fraction prevents us from applying any reconstruction model and, consequently, from obtaining reliable emission estimates.

[Figure]

**Figure A1. Comparison of the voltage measurements from three types of TGS included on chamber A. Upper plot shows the reference CH$_4$ observations measured from the reference instrument. Lower plot shows the voltage observations from TGS 2611-C00, 2600 and 2611-E00.**

**RC2:** **The writing and expression need substantial improvements. And many parts are very hard to follow. The manuscript needs to be polished by an experienced language editor, to thoroughly improve the fluency and remove grammar errors.**

The writing of the manuscript was thoroughly reviewed. As well the methods section was reorganised to present the information in a more logical manner. In the current methods section is structured as follows:

- Section 2.1: We introduce the TADI 2019 campaign.
- Section 2.2: We present the controlled releases and the sampling configuration.
- Section 2.3: We describe the low cost logger system and the meteorological data.
- Section 2.4: We present the preprocessing steps of the TGS data.
- Section 2.5: We describe the reconstruction of CH4 mole fraction from TGS voltage data and the metric used to evaluate the performance of reconstruction models.
- section 2.6: We explain the rationale in the selection of the training and testing sets.
- Section 2.7: We describe the inverse modelling framework used to estimate the release rates and locations.

**RC2:** **Discuss why E00 is bad compared with C00, e.g. in Fig.4 and 7.**

Our experiment has demonstrated that the TGS 2611-E00 sensor shows lower performance than the TGS 2611-C00 sensor in reconstructing CH4 mole fractions. This difference in performance was also observed and documented in a previous study (Rivera Martinez et al., 2022), where sensors measured artificial CH4 peaks under controlled conditions. In both experiments, we attribute the inferior performance of TGS 2611-E00 to the presence of a filter on top of the sensing material, designed to improve its selectivity to CH4 but with an effect on the sensor's sensitivity, acting as a low-pass filter.

The TGS 2611-E00 signal presents a lower amplitude compared to the TGS 2611-C00 signal, and a noticeable decay after each peak. This decay produces phase effects in the signal, which are also observed in the reconstructed CH4 mole fraction but absent in the TGS 2611-C00 signal. The lower performance is observed as a phase mismatch and a filtering of high-frequency features present in the measured peaks. Our suspicions regarding this performance difference were included in the discussion section with the following phrase:

The fast decay observed for reconstructed CH$_4$ mole fraction measurements after each voltage spike was attributed to the response time of the TGS sensor. The slow decay observed on Type E sensors was probably due to a filter integrated inside the sensor causing to improve CH$_4$ selectivity.

**RC2:** **Add designs, and photos on low-cost sensor instrument.**

The logger system used in this study is the same as the one used in our previous studies (Rivera Martinez et al. 2021, 2022). We have added the following sentence in section 2.3:

The logger system design was previously documented on Rivera Martinez et al. (2021) and Rivera Martinez et al. (2022).

**RC2:** **I suggest the authors provide spatial distributions of simulations and inversions for typical cases, e.g. to show the real emission sources and the inversed sources and their distances.**

The following figure, corresponding to the cost function, was added into the supplementary material to show the spatial distribution of the controlled release and the estimated location using the inversion framework.

[Figure]

Figure A15. Contour plot of the cost function for release #25 computed using assimilated gradients from TGS reconstructed data. The black and white stars show the location of the actual and estimated location respectively.

**RC2: I recommend the authors to make the inversion code publicly available to improve the wide influences and applications of this study.**

We have added the following text to the manuscript:

The codes developed in the frame of the Chaire Indutrielle TRACE ANR-17-CHIN-0004-01. They are accessible upon request to the corresponding author.

**RC2: Add regression coefficients (slope, intercept and p value) in all related figures (e.g. Fig.5-6; Fig. A4-A12) that are statistically significant.**

The figures were updated adding the $R^2$ and the p-value.

**RC2: Figure 4: Add scatter plots (and coefficients) of the corrected and reference data.**

The figure was updated. Currently it corresponds to figure 3.

[Figure]

Figure 3. Comparison of the voltage signal for one release (#8) from Chamber A before (Uncorrected) and after (Corrected) the baseline correction on (b) TGS 2611-C00 and (d) TGS 2611-E00, on which it is appreciated the correction of the offset preserving the amplitude enhancements linked to $CH_4$ variations. Scatter plot of the corrected (orange) and uncorrected (red) signal vs the reference CH4 observations for (c) TGS 2611-C00 and (e) TGS 2611-E00. (a) Reference $CH_4$ mole fractions, also corrected using the spike correction algorithm.

**RC2:** **Line99: participate in;**

Section 2.1 was updated to improve clarity:

In October 2019, TotalEnergies® performed multiple controlled releases at the TotalEnergies Anomaly Detection Initiative (TADI) facility, to investigate the capability of different detection and quantification techniques of $CH_4$ emissions from industrial facilities. The TADI test site is located northwest of Pau, France, with an approximate area of 200 m². It is equipped with infrastructure typical of oil and gas facilities (pipes, valves, tanks, etc) to simulate 'realistic' leaks. The terrain is flat but includes different obstacles that can affect the dispersion of the gases released to the atmosphere. Our experiment consisted of 41 controlled releases of $CH_4$ and $CO_2$, covering a wide range of emission rates of between 0.15 and 150 g $CH_4$ s⁻¹, with durations ranging between 25 to 75 minutes. We participated in this experiment to develop and test inverse modelling frameworks within the TRAcking Carbon Emissions (TRACE, https://trace.lsce.ipsl.fr/) program for the estimation of emission location and rates based on $CH_4$ mole fractions from high precision instruments (Kumar et al., 2022). We presented the inversion results for 26 releases from single point sources based on two inversion approaches, one relying on fixed-point measurements, and the other one on mobile near-surface measurements (the latter had already been documented in Kumar et al. (2021)). In both cases,

the emission estimates relied on CH$_4$ mole fractions from high precision instruments, and on a Gaussian plume model to simulate the local atmospheric dispersion of CH$_4$. The results from Kumar et al. (2022) for point source emissions yielded an emission rate error of between ~23 to ~30 % and a localisation error (within a 40 m × 50 m area) of between 8 and 10 m. The controlled releases were emitted from heights of between 0.1 m and 6 m above ground level, and inside the 40 m × 50 m ATEX (ATmospheres EXplosibles) zone of the TADI facility (see Fig. 1).

RC2: Line100-101: ambiguous for "for the estimation of … based on …high precision", better to separate this for another sentence? "And the TRACE program is …";

See answer for comment 3.

RC2: L112: consist of doing is better to be changed to consist of sth.

See answer for comment 3.

RC2: L115: You may mean "connected to an upstream chamber which holds the high precision instruments…"

See answer for comment 3.

RC2: L116: (Picarro  or LGR), or provide specific type;

See answer for comment 3.

RC2: L122: better to use the datasheet parameter: "less than 3 ppb per month"

We have update the phrase:

"In a previous study by Yver-Kwok et al. (2015), it was proven that these CRDS gas analysers ensure high precision measurements and a low drift over time, of less than one ppb per month, although the datasheet specifies a drift of 3 ppb per month (Picarro, 2017)."

RC2: L124-127: hard to follow, needs to be rewritten in short sentences;

See answer for comment 3.

RC2: L129: redundant, combine sensors: "the CH4 and environmental sensors"

Sentence corrected:

Table A6 shows the TGS and environmental sensors in each chamber, as well as the type of chamber.

RC2: L131: two  other sensors;

Updated sentence:

Each chamber contained at least three TGS units with voltage measurements sensitive to CH$_4$ and two other sensors measuring relative humidity/temperature and pressure/temperature.

RC2: L134-135: add "a"…ADC board ;… change "recorded" to records

Updated the sentence:

An AB Electronics PiPlus ADC board mounted on a Raspberry Pi 3B+ recorded the voltage drop across the load resistor, providing observations every 2 s.

RC2: L146: Why they are used in the training of models?

To reconstruct CH4 mole fraction from observed voltage from the TGS sensors we employed a data driven approach consisting in minimising the error between the predicted output and the reference measurement on an iterative process, using a multi layer perceptron or a 2nd degree polynomial. This approach requires a sufficient number of examples from which the model can learn the relationships between voltage variations and CH4 concentration. Contrary to the case of inverse modelling, it is not affected by the wind conditions since reconstruction models do not use wind information to derive CH4 mole fractions.

RC2: L217: We use

Sentence corrected:

To assess the performance of the reconstruction models to provide dry CH$_4$ mole fractions enhancements (above the background) from voltage drop measurements corresponding to the

TGS sensors, we used a normalised root mean square error (NRMSE) per release, weighted by the inverse of the maximum peak present in the release

**RC2: L219 and 222: presented**

The text was corrected:

where $y_i$ are the $CH_4$ mole fraction measurements provided by the high precision instrument, $\hat{y}_i$ are the reconstructed $CH_4$ mole fractions, n is the number of observations present in each release, and $h_{max}$ is the amplitude of the maximum mole fraction peak enhancement present in the release after removing the background.

**RC2: L220: the unit of hmax is ppm? And thus the NRMSE is dimensionless?**

Yes, the NRMSE is dimensionless. The figures and the text was updated accordingly.

**RC2: L244: change "are" to "were"**

The sentence was corrected:

The reconstruction models were trained and tested only once per chamber, following the distribution of the releases from Table 2.

**RC2: L245: Table 4**

Table 1 and 2 were moved to SM. Table 4 was updated to Table 2 (see previous comment).

**RC2: L275-276: how long is the typical time decay?**

The time delay from synchronisation between high-precision gas analysers and TGS chambers varies between 2 to 3 minutes depending on the day of the campaign.

**RC2: L316: discuss a bit on why**

The following text was added to the discussion section.

The combination of both sensors as input produced a reconstruction of CH4 mole fractions similar to using only one of the sensors (TGS 2611-C00). This can be explained by the fact that both of the TGS signals are highly correlated and do not add more information to the model, and the phase mismatch between both input signals produced by the filter on TGS 2611-E00 sensor.

**RC2: L333-335: redundant and a bit ambiguous**

The phrase was corrected to reduce redundancy and ambiguity:

For most of the cases, the modelled gradients assimilating the TGS data are closer to those assimilating the reference data than to the observed TGS data.

**RC2: L340, 348: comply with the journal requirements on capitals of figures and keep consistency through the text (Figure A14 and fig 9a).**

The manuscript was reviewed to comply with the journal requirements.

**RC2: L366-372: These contents seems to be more suitable for conclusion**

The paragraph was moved to the conclusion section.

**RC2: L431-433: The study of how many sensors are needed and the layout of these sensors are also needed.**

We have added the following text in the discussion section about the density of the network:

Density of sensor network

In our campaign, we deployed 7 chambers connected to air inlets placed on tripods at distances of between 40 and 50 m from the emission source to capture methane plumes under various conditions. Table 3 details the number of sensors used for emission flux estimations across the controlled releases.

The optimal number of sensors for emission flux localisation and estimation is complex, influenced by varying emission rates, environmental conditions, and setup configurations. Notably, when examining cases with uniform emission rates (1 g CH4/s), such as releases 12, 2, and 21 (with 3, 4 and 5 chambers respectively), a configuration of 4 to 5 sensors consistently produced the lowest errors for both sensor types. Yet, release 21 demonstrated that even five sensors may not guarantee low errors if the plume capture is suboptimal due to environmental factors or sensor placement.

We can contrast our setup with Riddick et al. (2022), who used four sensors approximately 30 m away from the source, but without detailing their individual contributions to emission calculations.

The optimal configuration of such a relatively dense network necessitates a thorough investigation, possibly through simulations of typical emissions and the strategic addition or removal of sensors to assess their impact. However, a comprehensive analysis of optimal network configuration was beyond the scope of our study due to the limited number of data points recorded.